# A contrast in sea ice drift and deformation between winter and spring of 2019 in the Antarctic marginal ice zone

Ashleigh Womack[1], Alberto Alberello[3], Marc de Vos[4], Alessandro Toffoli[5], Robyn Verrinder[6,2], Marcello Vichi[1,2]

[1]Department of Oceanography, University of Cape Town, Rondebosch, South Africa
[2]Marine and Antarctic Research centre for Innovation and Sustainability, University of Cape Town, Rondebosch, South Africa
[3]School of Mathematics, University of East Anglia, Norwich, United Kingdom
[4]Marine Research Unit, South African Weather Service, Cape Town, South Africa
[5]Department of Infrastructure Engineering, The University of Melbourne, Parkville, Australia
[6]Department of Electrical and Electronic Engineering, University of Cape Town, Rondebosch, South Africa

*Correspondence to*: Ashleigh Womack (ashleighwomack@gmail.com) and Marcello Vichi (marcello.vichi@uct.ac.za)

**Abstract.** Two ensembles of buoys, deployed in the marginal ice zone (MIZ) of the north-eastern Weddell Sea region of the Southern Ocean, are analysed to characterise the dynamics driving sea ice drift and deformation during the winter-growth and the spring-retreat seasons of 2019. The results show that although the two buoy arrays were deployed within the same region of ice-covered ocean, their trajectory patterns were vastly different. This indicates a varied response of sea ice in each season to the local winds and currents. Analyses of the winter data showed that the Antarctic Circumpolar Current modulated the drift near the sea ice edge. This led to a highly energetic and mobile ice cover, characterised by free-drift conditions. The resulting drift and deformation were primarily driven by large-scale atmospheric forcing, with negligible contributions due to the wind-forced inertial response. For this highly advective coupled ice-ocean system, ice drift and deformation linearly depended on atmospheric forcing. We also highlight the limits of commercial floating ice velocity profilers in this regime since they may bias the estimates of sea ice drift and the ice type detection. On the other hand, the spring drift was governed by the inertial response as increased air temperatures caused the ice cover to melt and break up, promoting a counterintuitively less wind-driven ice-ocean system that was more dominated by inertial oscillations. In fact, the deformation spectra indicate a strong de-coupling to large-scale atmospheric forcing. Further analyses, extended to include the deformation datasets from different regions around Antarctica, indicate that, for similar spatial scales, the magnitude of deformation vary between seasons, regions and the proximity to the sea ice edge and the coastline. This implies the need to develop rheology descriptions that are aware of the ice types in the different regions and seasons to better represent sea ice dynamics in the MIZ.

## 1 Introduction

Antarctic sea ice forms a natural barrier between the atmosphere and the Southern Ocean, modulating the exchange of heat, gases, and momentum, and contributing to the global climate system balances (Mcphee et al., 1987; Kohout et al., 2020). The seasonal sea ice zone undergoes one of the largest annual changes on Earth (Allison, 1997; Massom and Stammerjohn, 2010), with $\approx 15 \times 10^6$ km$^2$ of ice that forms and subsequently melts each year (Eayrs et al., 2019). During the winter advance season, the ice cover is characterised by the formation of frazil ice and relatively free-floating pancake ice floes (Doble and Wadhams, 2006; Wadhams et al., 2018; Alberello et al., 2022), which form during wavy conditions (Meylan et al., 2014). The dynamics and thermodynamics of these roughly circular and mobile floes 1-10 m in diameter (Alberello et al., 2019; Alberello et al., 2022) control the evolution of the marginal ice zone (MIZ; Doble et al., 2003), i.e. the outer sea ice region, where the interactions between the atmosphere and ocean are most intense and variable (Strong et al., 2017; Wadhams, 1986). The MIZ extent is primarily limited by the Antarctic Circumpolar Current (ACC), which flows clockwise

around the Antarctic continent (Vihma et al., 1996), and acts as the northern boundary for seasonal ice formation. However, further into the MIZ, where the influence of open-ocean waves is reduced (Doble and Wadhmas, 2006), larger pancakes can freeze together (Shen and Ackley, 1991) and eventually consolidate to form a coherent ice cover (Weeks and Ackley, 1986). Despite this anticipated seasonal consolidation, large variability in sea ice concentration from space is observed at the monthly scale in regions of 100 % ice coverage (Vichi, 2022), which may increase heat loss from the ocean. During the spring retreat season, the surface heat balance changes, causing the consolidated ice to break up (Squire et al., 1995) and form floes and brash ice with a wide range of diameters. This creates positive feedback through the reduction in albedo and the presence of more open water, where waves can more freely propagate and break the ice (Kohout et al., 2014; Passerotti et al., 2022). This subsequently leads to the further melt and retreat of the seasonal ice cover.

On time scales of a day or more, sea ice moves in response to oceanic and atmospheric forcing (Thorndike and Colony, 1982; Alberello et al., 2020; Womack et al., 2022), and is modified by internal ice stresses, which depend on the characteristics of the ice cover such as ice thickness, concentration and strength (Heil et al., 2009; Heil et al., 2011). However, wind forcing has been regarded as the primary forcing mechanism of ice drift (Nansen, 1902; Allison, 1989; Vihma et al., 1996; Womack et al., 2022). The momentum transfer from the winds to the sea ice can be described by a linear ratio, the wind factor (Nakayama et al., 2012), with a turning angle between the ice drift and wind direction ranging between 0-30$^{\circ}$, negative in the Southern Hemisphere (Leppäranta, 2011). Generally, the wind factor is 2 % for pack-ice conditions (Leppäranta, 2011), although larger values have been reported for pancake conditions in both the Arctic (e.g. Wilkinson and Wadhams, 2003; Lund et al., 2018;) and the Antarctic (e.g. Alberello et al., 2020; Womack et al., 2022). On shorter time scales, the inertia of sea ice becomes more important, and drift trajectories often include elliptical loops – inertial oscillations – superimposed on an approximately steady translation (McPhee, 1988). This is because sea ice, simultaneous with the ocean, responds to rapid changes in wind stress (Lei et al., 2021), such as the passage of cyclones (Hibler et al., 2006; Lammert et al., 2009; Gimbert et al., 2012a, b), at both the low (synoptic) frequencies to high (sub-daily) frequencies (periods; McPhee, 1988). However, unlike tidal forcing, the inertial response of sea ice has no direct high-frequency equivalent in neither the oceanic nor the atmospheric spectra (Heil and Hibler, 2002). Rather, a cascade of energy from the low frequencies to high frequencies, within the wind spectra, is required to generate the inertial-frequency power in the ice drift and deformation (Heil et al., 2009). This cascade arises from non-linear ice dynamics, as ice drift transfers its kinetic energy to the underlying ocean (Leppäranta, 2011). From this, high frequencies can be fed in back into the ice drift, and become trapped close to the semi-diurnal frequencies (Heil et al., 2009).

The ice cover is also highly deformable, as a result of the differential drift of individual floes (Hibler, 1974; Geiger et al., 1998; Girard et al., 2009). Sea ice deformation is a dynamic phenomenon that is constrained in space and time (Oikkonen et al., 2017; Rampal et al., 2019), but little is known about the heterogeneous Antarctic sea ice. During the last few decades, several methods have been developed to quantify sea ice deformation. The most common approach to determine deformation is computed from the strain rates – divergence, shear and total deformation (Lindsay, 2002; Leppäranta, 2011) – which are derived from the spatial gradients in the ice velocity field, at the vertices of polygonal buoy arrays (Stern and Lindsay, 2002; Hutchings and Hibler, 2008; Itkin et al., 2017; Aksamit et al., 2023). However, this leads to large uncertainties, since not all buoy arrays are deployed sufficiently well to determine deformation in this manner (Rampal et al., 2009) and the patterns align unfavourably in the MIZ (de Vos et al., 2022). In lieu of this, alternative approaches that do not depend on buoy geometry and orientation can be useful, even if to merely determine and differentiate dynamic regions (Aksamit et al., 2023).

Lagrangian dispersion statistics are conventionally used to characterize paths and structures in atmospheric and oceanic dynamical phenomena in order to identify topological and dynamical features within a flow field (LaCasce, 2008; Lukovich et al., 2017). The approaches can be sub-divided into single- (or absolute) and multi-particles methods. However, both single- and multi-particle statistics are needed for a full description of ice floe evolution (LaCasce, 2008). Lagrangian dispersion statistics applied to ice-buoy trajectories have been used extensively to quantify ice drift and deformation in the Arctic (e.g. Rampal et al., 2008, 2009; Girard et al., 2009; Lukovich et al., 2011, 2015, 2017, 2021). However, to our knowledge, only absolute dispersion has been considered in the Antarctic by Womack et al. (2022), and only for a single-buoy trajectory. In general, absolute dispersion provides a signature of circulation and organised structure in the flow field (Lukovich et al., 2021). It also estimates the linear time dependence of the fluctuating velocity variance, characteristic of turbulent diffusion theory (Taylor, 1922). Variations in ice-fluctuating velocity statistics associated to turbulent diffusion are considered to be related to sea ice deformation and internal ice stresses (Rampal et al., 2009; Lukovich et al., 2017).

Relative (two-particle) dispersion characterises the deformation of sea ice by using the temporal evolution of the separation between two Lagrangian trackers in the ice (Rampal et al., 2009; Lukovich et al., 2017). Martin and Thorndike (1985) showed that, in a statistical sense, there are similarities between the dispersion properties of sea ice and the dispersion of particles in turbulent fluids, although the underlying physics may be different. By using the separation of buoy pairs as a proxy of the combined strain-rate components, sea ice relative dispersion is a way to analyse deformation (Rampal et al., 2008), making it a valuable and important measure of sea ice dynamics.

Another alternative approach to studying sea ice dynamics is through the trajectory-stretching exponents (TSEs; Haller et al., 2021, 2022) – single-buoy stretching diagnostics. For a trajectory, TSEs are quasi-objective Lagrangian metrics of material stretching (Aksamit et al., 2023). Specifically, they estimate the true material stretching in gradually varying flows, and recently have been extended to approximate deformation of the sea ice cover (e.g. Aksamit et al., 2023). By resolving Lagrangian coherent structures, it allows for the identification of large sea ice variation, for example during cyclonic events or springtime ice melt. Large values correspond to periods and areas of significant convergence, divergence and shear (Aksamit et al., 2023).

Synoptic events have a significant influence on the evolution of Antarctic sea ice. However, our current understanding of the interactions between cyclones and sea ice remains limited (Vichi et al., 2019). The majority of the existing literature on extra-tropical cyclones does not consider what happens when they cross the MIZ. Additionally, very little field data of metocean (meteorological and oceanographic) conditions are available in the Southern Ocean, and even less in the MIZ (Derkani et al., 2020). This has had negative feedbacks on the remote sensing and prediction model network, which cannot depend on adequate ground truth to be validated with high confidence (Derkani et al., 2021; Lavergne and Down, 2022). Additionally, this has led to sea ice dynamics in the Antarctic MIZ being poorly understood – specifically ice drift, deformation and type. Womack et al. (2022) previously attempted to overcome these knowledge gaps by showcasing one of the longest ice-tethered buoy trajectories in the Antarctic MIZ, as it drifted for four months spanning winter and spring within the Indian Ocean sector, and under the influence of several synoptic cyclones. They demonstrated that wind forcing had a dominant physical control on ice drift, with the persistence of free-drift conditions within regions of 80-100 % remotely sensed ice concentration and > 200 km from the ice edge. Moreover, the drift was characterised by a strong inertial signature at ≈ 13.5 hours, which appeared initiated by passing cyclones. This implied and further corroborated that the concentration-based definition (15-80 %) is inadequate to define the MIZ and its composition, since the highly dynamic nature of the MIZ is maintained despite the high sea ice concentrations (> 80 %) observed from remote sensing products.

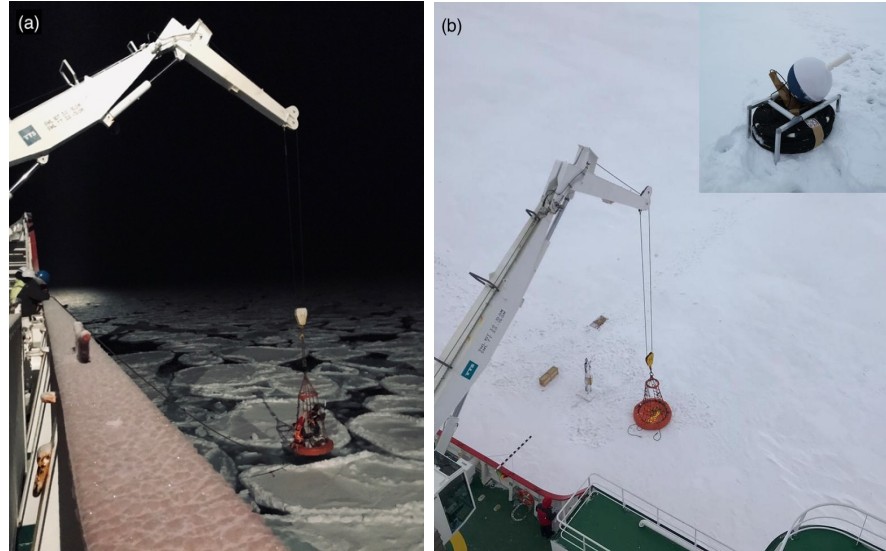

**Figure 1.** The sea ice conditions and deployment of a buoy during **(a)** the 2019 Winter cruise, on a pancake ice floe, using the ship's crane, and **(b)** the 2019 Spring cruise on consolidated ice floes. The inset in **(b)** depicts the frames specifically designed around the standard ISVPs for the 2019 Spring cruise. The diameter of the basket is 1.5 m. The inset is credited to Mardene de Villiers.

In this study, we extend the work of Womack et al. (2022) to two arrays of ice-buoys deployed during austral winter and spring in the north-eastern Weddell Sea region of the Southern Ocean, to provide information on sea ice dynamics at the synoptic and shorter time scales over two seasons. We correlate the in situ drift measurements with atmospheric reanalysis data to investigate the effects of extra-tropical cyclones on ice drift during each season, through the momentum transfer from winds and the generation of inertial oscillations of the sea ice. We also explore the differential drift between the buoys in each season to estimate the ice deformation rates and compare them with the few existing datasets from other regions, in search for general patterns of Antarctic sea ice deformation.

## 2 In situ observations and environmental data

In 2019, winter and spring scientific research expeditions aboard the SA Agulhas II were conducted along the Good-Hope line ($0^o$ E) to the Antarctic MIZ as part of the *Southern oCean seAsonal Experiment* (SCALE; Ryan-Keogh and Vichi, 2022). These expeditions generated a large set of comprehensive data from the physics to the biogeochemistry of the MIZ in the Atlantic sector. In this study, we focus on the analysis of seven GPS-tracked ice buoys deployed during these two expeditions, which supplied the sub-daily and true daily information on Lagrangian ice drift and deformation that are needed to validate satellite ice drift products (Heil et al., 2001, 2009).

The Winter cruise departed from Cape Town on 18 July 2019 and reached the MIZ on 26 July, at $56.5^o$ S and $0.1^o$ E. Three polar Iridium Surface Velocity Profilers (MetOcean model; de Vos et al., 2022; de Vos et al., 2023), hereafter simply referred to as ISVP 1-3 were deployed during pancake ice conditions (1-5 m floes, and $\approx$ 40 cm thick; Fig. 1a), along the $0^o$ meridian. Features of the sampling sites have been previously detailed in Skatulla et al. (2022). ISVP 1, was initially deployed in water, in between pancake floes, while ISVP 2 and ISVP 3 were deployed directly onto large pancake ice floes. These ISVPs were expendable devices that recorded position, air and ice temperature, and barometric pressure. Only their GPS positions are used in this analysis due to reliability issues of the meteorological data. Details of the deployments and lifetimes of the buoys are given in Table 1.

The Spring cruise departed from Cape Town on 11 October 2019 and was the first time that the Good Hope line was sampled during austral spring. The ship entered the MIZ on 20 October at 55.0$^{o}$ S and 0.0$^{o}$ E and the first standard Iridium Surface Velocity Profiler (Pacific Gyre model; de Vos et al., 2022, de Vos et al., 2023), denoted ISVP 4, was deployed on 24 October along the 0$^{o}$ meridian (for ease of reading, all the drifting buoys are named ISVP and numbered sequentially). During this cruise, specifically designed frames were built around these buoys to allow them to stand firmly on the ice floes without damaging the non-polar battery (inset in Fig. 1b), and to ensure that they would operate as Lagrangian-ice trackers. The other three buoys – ISVP 5, ISVP 6 and an ice-tethered, non-floating Trident Sensors Helix Beacon (Womack et al., 2022; Womack et al., 2023) – were later deployed more than 5$^{o}$ (> 400 km) east of ISVP 4. All four of these devices were deployed during first-year ice conditions (> 45 cm thick, Fig. 1b), and features of these sampling sites have been detailed in Johnson et al. (2023). The standard ISVPs recorded position, air temperature and barometric pressure, while the Trident recorded position and air temperature. Similar to the winter deployments, only the GPS positions are used in the analysis. Details of the buoys' deployment and lifetime can be found in Table 1.

**Table 1:** Operational details of the Winter and Spring 2019 buoys

| | Deployment date (2019) | Deployment position | Sampling frequency | Analysis end date (2019) | Number of analysed days | Total drift distance (km) |
|---|---|---|---|---|---|---|
| **Winter** | | | | | | |
| ISVP 1 | 27 July at 01:00 | 57.05$^{o}$ S, 0.10$^{o}$ W | 30 minutes | 5 October at 12:00 | 70 (10*) | 2642.85 |
| ISVP 2 | 28 July at 05:07 | 57.17$^{o}$ S, 0.00$^{o}$ E | 1 hour | 25 August at 03:00 | 28 | 1036.97 |
| ISVP 3 | 27 July at 17:09 | 57.92$^{o}$ S, 0.02$^{o}$ W | 1 hour | 5 October at 12:00 | 70 | 2552.28 |
| **Spring** | | | | | | |
| ISVP 4 | 24 October at 12:00 | 59.33$^{o}$ S, 0.06$^{o}$ E | 1 hour | 7 December at 12:00 | 44 | 1101.89 |
| ISVP 5 | 28 October at 10:00 | 59.35$^{o}$ S, 6.57$^{o}$ E | 1 hour | 9 December at 12:00 | 42 | 992.44 |
| ISVP 6 | 29 October at 12:00 | 59.37$^{o}$ S, 8.16$^{o}$ E | 1 hour | 7 December at 12:00 | 39 | 940.99 |
| Trident | 30 October at 12:00 | 59.47$^{o}$ S, 10.89$^{o}$ E | 4 hours | 2 December at 00:00 | 33 | 650.73 |

*ISVP 1 left the AMSR2-derived ice edge within 10 days.

The in situ observations were integrated with environmental data retrieved from satellite and reanalysis products. Larger-scale meteorological conditions in the form of mean sea level pressure (mslp), 10 m wind velocity, and 2 m air temperature were obtained from ERA5 (Copernicus Climate Change Service (C3S), 2017), with an hourly time interval. Prior literature validated this reanalysis product in the Weddell Sea (King et al., 2022) and at ≈ 30°E (Vichi et al., 2019). These variables were then bi-linearly interpolated in space to the buoys' locations, hourly for all ISVPs and four-hourly for the Trident, to ascertain the synoptic atmospheric forcing during both seasons. Sea ice concentration (SIC) data at 3.125 km spatial resolution were acquired from the passive microwave Advanced Microwave Scanning Radiometer 2 (AMSR2) sensor (Spreen et al., 2008), and complemented by the 25 km spatial resolution Special Sensor Microwave Imager/Sounder (SSMIS) product (NSIDC, 2023).

In addition to their low temporal (and spatial) resolution, the use of satellite products is known to be limited in their application to the broad Antarctic MIZ, where ice type is less related to the concentration value (Alberello et al., 2019; Vichi, 2022). The amount of in situ data available is yet not sufficient enough to be used for regular validation (Aaboe et al., 2021). This has had drawback effects for prediction models, which are impaired by significant biases in the Southern Ocean (e.g. Yuan, 2004; Li et al., 2013; Zieger et al., 2015). Therefore, in this study we do not use SIC other than to estimate the ice edge, and hence, have rather inferred ice type from the buoys' drift features and wind response (see Section 5.1).

Herein, the 0 % SIC is used to define the sea ice edge rather than the conventional 15 % SIC, even though this 0 % SIC region has been recognised as being heterogeneous and fragmented. This is because satellite product algorithms tend to underestimate the SIC in thin ice as well as close to the ice edge (Pang et al., 2018), where ice melt and growth conditions make up a large component of the sea ice regime (Agnew and Howell, 2003). Furthermore, Womack et al. (2022) reported that their results, of the buoy's distance from the ice edge, in the Antarctic were only marginally affected by this choice, with a maximum difference between the 0 % and 15 % SIC of less than 50 km. This is still within the range of differences between satellite products.

By construction, ISVPs continue to drift in the ocean after ice melting and can be further refrozen in between floes. Therefore, there is uncertainty on whether these buoys remained within the MIZ during their drift, especially for ISVP 1 as it was deployed in between ice floes. Using daily SIC data to estimate the ice edge, we determined the dates when the ISVPs left the ice cover. During winter, ISVP 1 first left the AMRS2 ice cover 10 days after its deployment (see Fig. S1 of the Supplement). ISVP 3, initially located the furthest south and thus further from the ice edge, on the other hand remained within the AMSR2 ice cover the longest (until 20 September 2019). ISVP 1 and ISVP 3, eventually both left the SSMIS ice edge on 5 October 2019. We subsequently ended our analysis on 5 October (Table 1), but caution was taken when analysing this dataset. ISVP 2 stopped transmitting on the 25 August, presumably due to battery failure. Similarly, the Spring ISVPs also would have been able to function as open-ocean drifters after the ice melted. Thus, we estimated that they left the ice cover between 7-9 December 2019 using the AMSR2 and SSMIS 0 % ice concentrations (see Fig. S2 in the supplementary material). The non-floating Trident buoy sank on 2 December 2019 due to ice melting.

## 3 Methods

### 3.1 Extra-tropical cyclone identification

Since polar cyclones in the Southern Ocean typically occur every five to seven days (Hoskins and Hodges, 2005; Vichi et al., 2019; and references therein), and due to the fact that the Winter and Spring buoys' trajectories are confined, the cyclones can be tracked without the need of an automatic tracking algorithm. Following the procedure by Womack et al. (2022), a visual inspection method of the ERA5 mslp and 2 m air temperature fields at four-hourly intervals is applied to investigate the time window at which the cyclone cores were closest to the buoys' location. Wei et al. (2013) reported that between 1979 to 2013, the mean intensity of cyclones in the Southern Ocean was 967.4 hPa during winter, 968.4 hPa during spring, 972.4 hPa during summer and 968.7 hPa during autumn. Therefore, for our analysis we only consider cyclones with core pressures < 970 hPa. Eight cyclones for winter and seven for spring have been identified by low-pressure troughs < 1000 km from the buoys, and by an increase in air temperature to near melting point on the eastern flank of the cyclones (Vichi et al., 2019). The dates when these cyclones were closest to the buoys have been computed using a nearest-neighbour method, and later associated with the ice drift and dispersion analyses.

### 3.2 Buoy kinematic parameters

The GPS position of all seven buoys was communicated via the Iridium system, with an accuracy of < 5 m. Since some data were irregular, missing or had duplicates, the position data were interpolated to a regular interval of one hour for all six ISVPs and four-hourly for the Trident. The higher temporal resolution of the ISVPs was kept since it provided more accurate data on ice drift. However, the following methods were also repeated for the Spring ISVPs using the four-hourly time interval and the difference was negligible. For each of the buoys, their latitude and longitude positions can be used to derive their downwind zonal ($u$) and meridional ($v$) components using the standard linear approximation:

$$u = \frac{\Delta x}{\Delta t}, \tag{1}$$

$$v = \frac{\Delta y}{\Delta t}, \tag{2}$$

where $\Delta x$ and $\Delta y$ are the zonal and meridional geodesic distances travelled between points along each buoys' trajectory, at time interval $\Delta t$. The speed is taken as the magnitude of the resultant of the velocity components.

For buoys, errors in drift measurements depend on the accuracy of position and time readings. Errors due to the timing of GPS position measurements are generally quite small (Dierking et al., 2020). Hutchings and Hibler (2008) reported that velocity errors are < 10 % for sampling intervals > 1 hour, and therefore it can be neglected (Dierking et al., 2020). The position error estimation can be attributed to the tracking error between two consecutive GPS positions, and the errors in the GPS reference points (Dierking et al., 2020), denoted by Lindsay and Stern (2003) as the geolocation error. The tracking error for buoys is however zero since the buoys remained fixed relative to the ice floe on which they were deployed. The geolocation error for these buoys is also taken to be negligible as the buoys drifted with little latitudinal change, where we can assume identical geolocation errors, which would cancel out when calculating the drift velocity and speed.

A fast Fourier transform is applied to the buoys' velocity time series to derive the power spectral density as in Heil et al. (2009). As the buoy data were in the form of a discrete time series, a Hamming window in the time domain is used to minimize frequency leakage (Heil et al., 2009; Glover et al., 2011). A Morlet wavelet analysis is additionally computed to examine how the frequency domain changed over time (Liu and Miller, 1996; Torrence and Compo, 1998; Womack et al., 2022). To better detect inertial oscillations in the wavelet analysis, a high-pass Butterworth filter is also applied with a cut-off threshold of one day.

Both buoy arrays were deployed and drifted in the deep Southern Ocean away from the Antarctic continent. Therefore, we do not consider tidal forcing, which is known to be negligible in off-shore locations (Heil et al., 2009; Lei et al., 2021; Alberello et al, 2020). Moreover, these buoys were near an amphidromic point (Lu et al., 2021; their Figure 10), which is a geographical location of the ocean where the main tidal fluctuation (M2) is negligible – ranging between 20-60 cm at most (Martine and Dalrymple, 1994; Kamphius, 2000).

The meander coefficient (M) is computed to assess the effective translation associated with the buoys' drift. This is defined as the ratio of the total accumulated distances travelled by a buoy (I) to net geodesic displacement $\Delta D$ (Vihma et al., 1996; Heil et al., 2009; Heil et al., 2011):

$$M = \frac{I}{\Delta D}. \tag{3}$$

This is first analysed as a time series for each time step, showing the cumulative change as the time window increased. It must be noted that $M$ is a function of time over which it is computed, and on the sampling intervals transmitted by the buoys (Heil et al., 2009). For this reason, we also compute a daily discrete meander coefficient to highlight deviations at the synoptic scale.

### 3.3 Wind factor and ocean current's drift

To examine the relationship between ice drift and surface winds, we adopt the linear relation described by Thorndike and Colony (1982). This method relates the drifting buoy velocity ($u_i$, $v_i$) and ERA5 10 m wind velocity ($U_{10}$, $V_{10}$) as follows:

$$\begin{bmatrix} u_i \\ v_i \end{bmatrix} = F \begin{bmatrix} \cos\theta & \sin\theta \\ -\sin\theta & \cos\theta \end{bmatrix} \begin{bmatrix} U_{10} \\ V_{10} \end{bmatrix} + \begin{bmatrix} \overline{c_u} \\ \overline{c_v} \end{bmatrix}, \tag{4}$$

where $F$ is the wind factor, $\theta$ is the turning angle and $\overline{c_u}$, $\overline{c_v}$ represent the mean (residual) ocean currents over the analysed period. The counter-clockwise rotation matrix is applied as the angle $\theta$ is negative (left deflection) in the Southern Hemisphere. In Eq. (4), the time variations of F, $\theta$, and $\overline{c_u}$, $\overline{c_v}$ are not considered, and thus may vary with the time period over which they are computed, and also the sampling interval of the transmission of the buoys. These constants are calculated using the least squares regression technique described by Kimura and Wakatsuchi (2000) and Kimura (2004) and are fully detailed in Womack et al. (2022), their Eq. (10-15).

The mean ocean currents are derived by subtracting the ERA5 wind-driven motion from the in situ ice motion – that is, the portion that is not linearly related to the variation of wind speed (Kimura, 2004). As such, they carry multiple uncertainties that are not directly quantifiable. To the best of the authors' knowledge, there are no robust methods to determine under-ice geostrophic currents from remote sensing platforms, and to extract proper statistics. Furthermore, there are currently

no in situ measurements for currents within the Antarctic MIZ region (e.g. Alberello et al., 2020). The only in situ observations of underlying ocean currents in the Antarctic were reported by Geiger et al. (1998). These observations were measured along the coast in the western Weddell Sea, where the Antarctic Coastal Current is located, and where land-fast and pack-ice conditions dominate. These ice types do not have an equivalent spatial morphology to the MIZ ice conditions (Vichi, 2022), and the coastal current incurs significantly different driving mechanisms. Consequently, while the residual currents cannot be validated by in situ data, we did compare them to ones from the Copernicus GlobCurrent database in the region of the Winter buoys' trajectories. This is fully detailed in Appendix B.

As previously done in Womack et al. (2022), their Eq. (16 and 17), we also compute the vector correlation coefficient $R_v^2$ and the Pearson correlation coefficient $R_{w,i}^2$ (with 95 % confidence interval) to estimate the fraction of the variance of the ice-drift velocity that is explained by the wind velocity, and the linear relationship between the magnitude of ice drift and wind speed.

### 3.4 Lagrangian measures for dispersion and deformation assessment

The absolute dispersion is used to characterise ice motion, and for an ice buoy in an ensemble of buoys, it is defined as (Taylor, 1922; Lukovich et al., 2017, 2021):

$$AD^2 = \langle |x_i(t) - x_i(0) - \langle x_i(t) - x_i(0) \rangle|^2 \rangle, \tag{5}$$

where $x_i$ is the zonal or meridional position of the $i$-th particle in the ensemble, as a function of the elapsed time $t$. We also applied the method by Lukovich et al. (2017) to calculate the relative dispersion of any two-buoy arrays:

$$RD^2 = \langle |x_i(t) - x_{i+1}(t) - \langle x_i(t) - x_{i+1}(t) \rangle|^2 \rangle, \tag{6}$$

which is defined for adjacent particle pairs $x_i$ and $x_{i+1}$ in the zonal or meridional direction. For both Eq. (5 and 6), the angular brackets denote the ensemble mean over the number of buoys (pairs) in the array. The totals are computed as the sum of the zonal and meridional components, for the time period when all buoys, in each season, transmitted together.

In this study, Green's theorem could not be applied to compute the strain rates i.e. three-particle dispersion. Both the Winter and Spring buoys were deployed in a quasi-linear buoy array geometry, and thus all triangles formed by the buoy positions had small angles (< 15°). This would have given unreliable calculations of the strain rates (Itkin et al., 2017), and the reduction in accuracy from a large array to only two buoys is unknown (Alberello et al., 2020). In lieu of this, we further analysed the dispersion of sea ice using the methods proposed by Rampal et al. (2008), which defines a proxy of the full strain rate tensor ($\dot{\varepsilon}_{tot}$). They considered two buoys, namely 1 and 2 with absolute positions $\vec{X}_1$ and $\vec{X}_2$ respectively, and with a separation $\vec{Y} = \vec{X}_2 - \vec{X}_1$. If these two buoys, initially separated by $L0 = ||\vec{Y}(0)||$, are observed after a time $t = \tau$, their separation changes to $l(\tau) = ||\vec{Y}(\tau)||$. The change in separation is then defined as:

$$\Delta r = ||\vec{Y}(\tau)|| - ||\vec{Y}(0)|| = l(\tau) - L0. \tag{7a}$$

$\Delta r$ is then computed as a function of $\tau$:

$$\Delta r = l(t + \tau) - l(t). \tag{7b}$$

In fluid mechanics, the dispersion process is characterised by the mean square change in separation $\langle \Delta r^2 \rangle$, while from a solid mechanics' perspective, there is a consensus that it is more relevant to express the dispersion in terms of a deformation rate ($\dot{D} = \frac{\Delta r}{\tau L0}$), using the standard deviation (Girard et al., 2009; Rampal et al., 2008, 2009; Weiss, 2013):

$$\sigma_{\dot{D}} = \langle \left( \frac{\Delta r}{\tau L0} - \langle \frac{\Delta r}{\tau L0} \rangle \right)^2 \rangle^{1/2}, \tag{8}$$

where angular brackets again denote the ensemble mean, computed over the number of buoy pairs in the cluster, for the time period when all buoys, in each season, transmitted together. Rampal et al. (2008) demonstrated in the Arctic that $\sigma_{\dot{D}}$ is proportional to $\dot{\varepsilon}_{tot}$. We remind that the $\sigma_{\dot{D}}$ diagnostics is not sensitive to solid rotations. It only quantifies the overall deformation due to divergence, convergence and/or shear, but does not allow to distinguish between them.

As an additional proxy for sea ice deformation, we applied the methods derived by Aksamit et al. (2023), and computed the TSEs for individual buoys in each season as:

$$\text{TSE}_{t_0}^{t_N}(\mathbf{X}_0) = \frac{1}{t_N - t_0} \log \frac{|\dot{\mathbf{X}}(t_N)|}{|\dot{\mathbf{X}}(t_0)|}, \tag{9}$$

where $\mathbf{X}_0$ is the initial position, $\dot{\mathbf{X}}$ is the derivative with respect to time (i.e. drift speed), and TSE is the measure of trajectory stretching or contraction (Haller et al., 2021) for discrete data (Aksamit et al., 2023). Similar to that of Lagrangian dispersion statistics, TSEs do not differentiate between contributions from divergence, convergence, or shear to the stretching of the sea ice. Instead, they note the stretching in the direction of vectors, tangent to the trajectory. TSEs are computed forward-in-time from the starting time $t_0$ and for each time step along the buoy's trajectory. The TSEs in each season were computed over a one-day integration period as to capture the daily-scale Lagrangian stretching, and to relate them to the drift kinematics and meander coefficient.

## 4 Results

In this section, all results from the multiple diagnostics in Section 3 are presented and explained. The diagnostics are considered complementary a priori, and they are comparatively discussed, analysed, and correlated in Section 5. The spatial and temporal changes in atmospheric conditions obtained from ERA5, in the vicinity of the buoys, and used throughout this section are fully described in Appendix A.

### 4.1 General drift and meandering

All seven buoys were initially deployed in the north-eastern Weddell Gyre region (Fig. 2). The Winter buoys experienced a significant eastwards transport, with ISVP 1 and ISVP 3 travelling over $25^\circ$ ($> 1500$ km) eastwards, with a smaller latitudinal variation of $\approx 4^\circ$ ($\approx 440$ km). ISVP 2 only travelled to $\approx 10^\circ$ E ($\approx 640$ km) since it stopped transmitting on 25 August 2019 (red triangle in Fig. 2a). These three trajectories were characterised by large sharp turns and meanders in response to the eight cyclones. The Spring buoys were deployed in the same region as the Winter buoys. However, the prevailing trajectory patterns of the Spring buoys were vastly different. The Spring buoys only drifted $3^\circ$-$7^\circ$ eastwards ($\approx$

190-400 km; Fig. 2b), but their meridional drift was significant and contributed to almost half of their total drift, as they travelled between $1^o$-$2^o$ ($\approx$ 100-200 km) almost directly northwards. Their trajectories also exhibited sharp turns and meanders in response to the seven spring cyclones. Additionally, during the prominent northwards drift, the Spring buoys were characterised by small cyclic loops, which are indicative of inertial oscillations (see Sect. 4.4).

385

The overall drift pattern of the buoys can be further described through the buoys' meander coefficients (Eq. 3). The final cumulative meander coefficients were < 1.5 for all the Winter buoys, indicating a predominantly straight trajectory. The low meander coefficient indicates that the drift of buoys was influenced by the ACC. Vihma et al. (1996) also reported a meander coefficient value of 1.4 in the region of the ACC. Conversely, the cumulative meander coefficients for the Spring

buoys ranged between 2.5 and 3, signifying a more oscillatory trajectory.

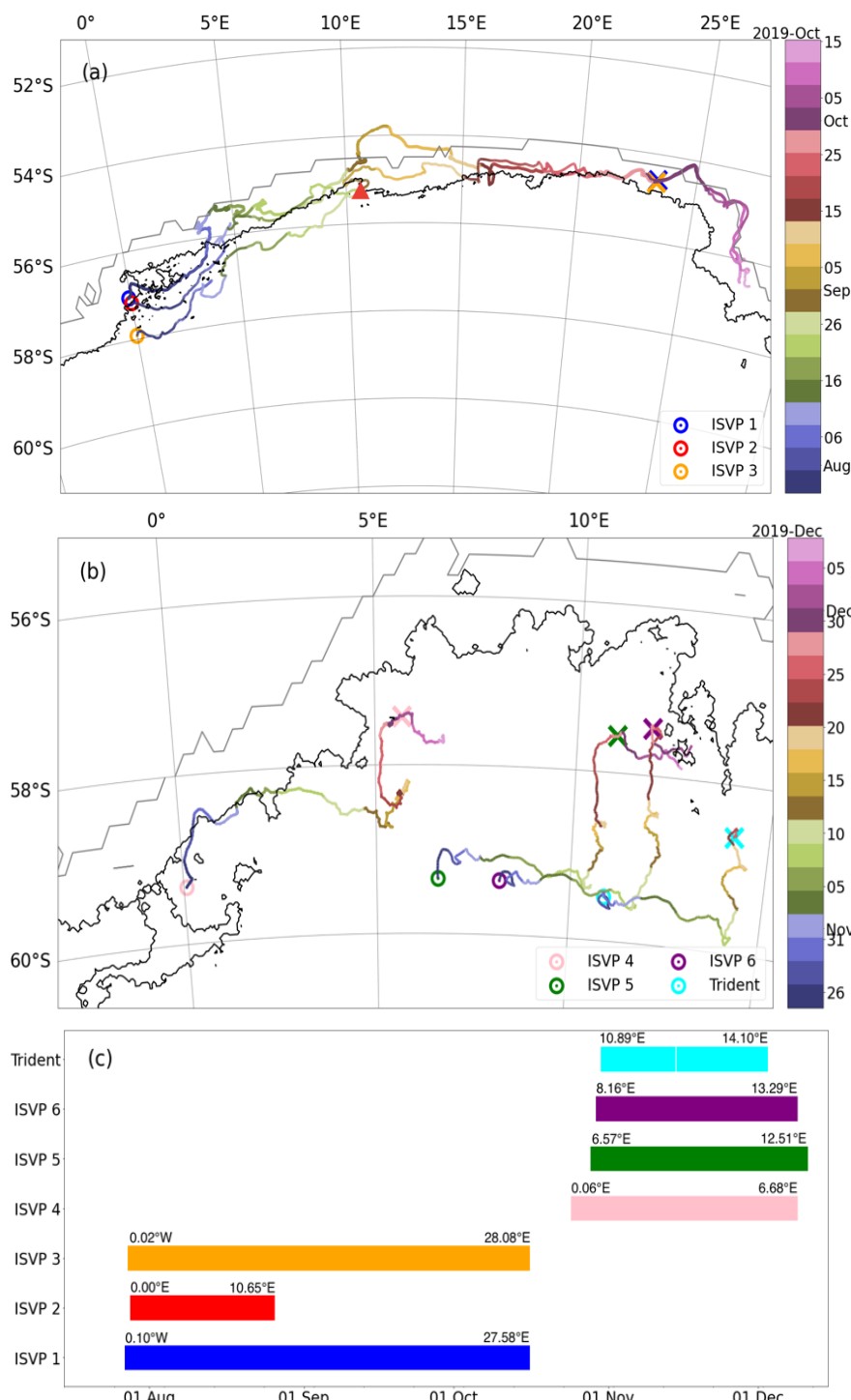

**Figure 2**. **(a)** Trajectories of the 2019 Winter buoys. The black (grey) contour denotes the AMSR2 (SSMIS) 0 % sea ice concentration on 30 September 2019 – the date of approximate austral sea ice maximum. The colour circles denote the start position of each buoy. The corresponding colour crosses of ISVP 1 and ISVP 3 denote their positions on 30 September 2019. The termination point of ISVP 2 (25 August 2019) is denoted by the red triangle. **(b)** Same as **(a)** but for the 2019 Spring buoys. The colour crosses and concentration contours are for 2 December 2019 – the day when the Trident buoy stopped transmitting data. **(c)** The operational periods of all buoys in this study, with corresponding colours to the time gradient maps. The deployment and final longitudes for each buoy are included above all operational periods.

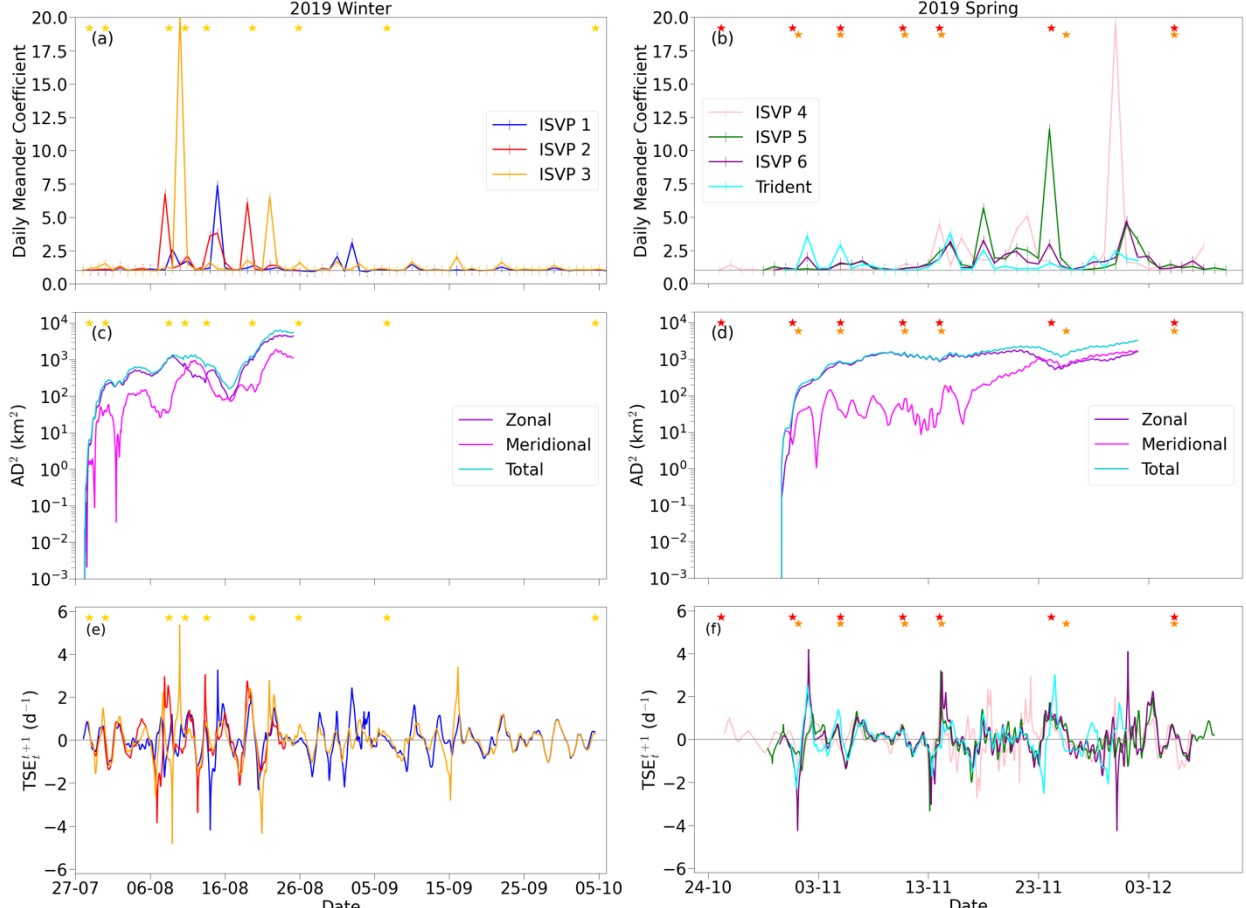

**Figure 3. (a-b)** Time series of the daily meander coefficient for the duration of the buoys' deployment, where the vertical markers are indicative of the daily time interval and the horizontal grey line at 1 denotes straight-line drift. **(c-d)** Time series of the absolute dispersion for the period when all buoys were transmitting together ($\approx$ 30 days for both seasons). **(e-f)** Time series of the TSEs for the duration of the buoys' deployment. The Winter buoys are shown on the left and the Spring buoys on the right. The star symbols are the same as in Fig. A1.

The time series of the daily meander coefficient varied between the two seasons (Fig. 3a and b). The highest values in winter occurred as a result of the second to the fifth cyclones (8-19 August), but rapidly decreased afterwards to fluctuate between 1-2. This reduction and the mostly straight-line drift of ISVP 1 and ISVP 3 can be seen in Fig. 2a, where from late August, their trajectories became predominantly eastwards with only small turns and deflections. However, ISVP 1 did exhibit increased meandering between 26 August and 2 September, which correlates with dates when ISVP 1 drifted in the open water outside of the AMSR2 estimated sea ice edge (Fig. S1 in the supplementary material). The largest meander coefficient values in spring occurred during two periods when inertial oscillations were present (see Fig. 2b). There were, however, two earlier increases in the meander coefficient particularly for the eastward cluster of the Spring buoys, during the second and third cyclones (31 October and 5 November, respectively). Since the meander coefficient is also affected by the sampling time step, the Spring ISVPs were therefore recomputed using the Trident's four-hourly time interval (Fig. S5a in the supplementary material). The magnitude and timing of the peaks remained the same, indicating that the relative motion in connection with the passage of the cyclones is realistic.

The evolution of the buoy trajectories is also described by the absolute dispersion (Fig. 3c-d) and the relative dispersion (Fig. S6 of the supplementary material) – from the time when all buoys in each season began transmitting together. For this reason, they both have been computed for $\approx$ 30 days using Eq. (5 and 6). During winter, the total absolute dispersion

(i.e. zonal and meridional components combined in quadrature) was predominantly influenced by the zonal displacement (Fig. 3c), while its corresponding relative dispersion was largely governed by the meridional separation (Fig. S6a). This was a result of these buoys being deployed meridionally along $0^o$ E (Fig. 2a) and drifting coherently eastwards with the winds and the ACC. However, both methods indicated substantial fluctuations of all components in response to the passage of cyclones, indicating a highly deformable ice cover. During spring, the total absolute dispersion and relative dispersion were both predominantly governed by their zonal component (Figs. 3d and S6b, respectively). However, when the Spring buoys switched to drift northwards (Fig. 2b), after the fourth cyclone (10 November), this led to the eventual dominance of the meridional displacement. Concurrently, the meridional separation exhibited greater changes in response to these cyclones. However, its magnitude was smaller and thus its contribution to the total was minor in comparison to the predominantly constant zonal separation. This indicates that the Spring buoys moved more coherently and as an aggregate despite the initial larger deployment distance. It is also noteworthy that there are minor fluctuations in all of the dispersion components during the spring season, which correspond to the dates of the cyclic loops and meanders in the buoys' trajectories (Figs. 2b and 3b). Overall, we however find the absolute dispersion is less sensitive to the spatial orientation of the deployments, and thus more useful to characterise the dominant dispersion direction.

During both seasons, increased local TSEs occurred with the passage of the cyclones, often before and/or after the cyclone core was closest to the buoys' locations (Fig. 3e and f). This is because cyclones push the ice edge southwards, as warm air is advected poleward on their eastern flank. This compression of the ice cover is then followed by the relaxation and northwards movement of the ice edge when southerly winds, on the western flank, prevail (Vichi et al., 2019). Although being conceptually different, the TSE and the meander coefficient show the same evolution. TSEs allow to further distinguish the compression and the stretching, with the latter always observed after the former, and the stretching events are generally linked to the increase of the meander coefficient. In winter, ISVP 1 and ISVP 3 indicated increased stretching outside of cyclonic activity (after 26 August), which is not visible neither in the meandering coefficient nor in the dispersion. Sea ice is still deformable in this period, likely due to oceanic influences (Aksamit et al., 2023) or changes in wind direction (Itkin et al., 2017) that are not directly related to cyclones. Also during spring, the buoys indicated increased and more erratic TSEs during periods of no cyclonic events (13-24 November and 2 December). These however, like their corresponding meander coefficients, coincided with the two periods of inertial oscillations (attended to in Sect. 4.4). We note that, like the meander coefficient, TSEs are influenced by the sampling frequency. Therefore, the Spring ISVPs were recomputed using the Trident's four-hourly time step (Fig. S5b in the supplementary material). Their magnitudes varied slightly but their time series continued to indicate the same phenomena in connection with the passage of the cyclones.

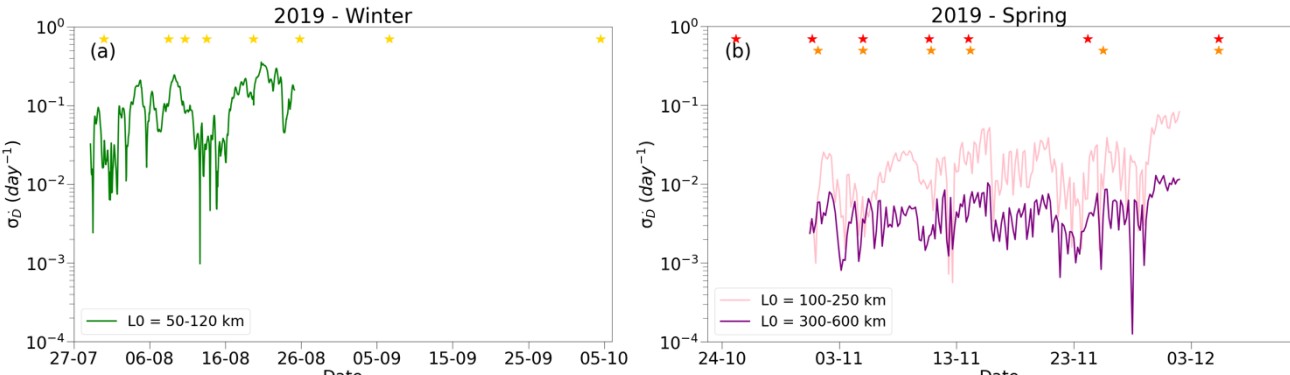

**Figure 4. (a)** Time series of the standard deviation $\sigma_{\dot{D}}$ of the deformation rate $\dot{D}$ for the Winter buoys (computed from the dispersion of buoy pairs for $\approx$ 30 days for both seasons). **(b)** Same as **(a)** but for the Spring buoys. L0 denotes the initial length scale of each group of buoys. The stars are the same as in Fig. A1.

**4.2 Deformation rate estimates**

The time series of the deformation rate $\sigma_{\dot{D}}$ for both seasons is shown in Fig. 4a and b. As ISVP 4 was deployed $\approx 5^{\circ}$ west of the other Spring buoys, we separated the analysis into two smaller clusters based off their initial length scales (L0 = 100-250 km and 300-600 km), as shown in Fig. 4b. This allowed for a more comprehensive analysis of sea ice deformation at different spatial scales, and not only between the two seasons.

The $\sigma_{\dot{D}}$ measured by the Winter buoys was not only larger than during spring, but it also exhibited greater fluctuations in relation to the passage of cyclones (Fig. 4a). The most notable decrease of the $\sigma_{\dot{D}}$ occurred during the passage of the second to the fourth cyclones (8-13 August) that came in close succession. As this proxy only quantifies the magnitude of the total deformation rate, we cannot discern whether this was predominantly due to divergence, convergence and/or shear. This did however occur in relation to the decrease in both the absolute (Fig. 3c) and relative dispersion (Fig. S6a), along with the large peaks in the meander coefficient (Fig. 3a), which is indicative of the compression of the ice cover by the cyclones. Overall, these large fluctuations of the $\sigma_{\dot{D}}$ indicate that the winter ice cover was deformable in relation to the changing winds at the scale of 50-120 km, allowing for opening of leads and rafting of floes.

Although the $\sigma_{\dot{D}}$ sampled by both spring spatial scales varied due to the passage of cyclones, it also exhibited regular and relatively uniform fluctuations throughout the analysed period that were not associated with the cyclones (Fig. 4b). Therefore, the deformation of the spring ice cover was less correlated to the cyclones and their changing winds, relative to during winter. Additionally, the $\sigma_{\dot{D}}$ from the larger cluster exhibited a more "flattened" trend in comparison to the smaller cluster. Thus, its time series exhibited a smaller $\sigma_{\dot{D}}$. This indicates that there was a smaller magnitude of deformation at these larger length scales.

**4.3 Metocean drivers of sea ice drift**

Figure 5 shows the zonal and meridional velocity components of ISVP 1 (a-b) and ISVP 3 (c-d), during winter, and ISVP 4 (e-f), during spring, compared to the co-located ERA5 wind-velocity components. The time series of all Winter and Spring buoys can be found in the supplementary material (Figs. S7 and S8, respectively). The overall drift velocity of the Winter buoys showed a good correlation to the wind (Table 2), suggesting low to absent internal stresses in the ice cover, which is typical of free-drift conditions. ISVP 1 showed large peaks (> 1 m s$^{-1}$) on 11 August and between 26 August and 5 September. These periods correlate with the dates when ISVP 1 drifted outside of the AMSR2 estimated sea ice edge (Fig. S1 in the supplementary material).

As a reference for the Spring buoys, ISVP 4 initially drifted similarly to the wind velocity (Fig. 5e and f), although with a dampened signal because these buoys were deployed on consolidated sea ice conditions. After the second spring cyclone (31 October), when air temperatures increased (Fig. A1b), ISVP 4's velocity fluctuations began to amplify. However, after the fourth cyclone (10 November) these fluctuations appear to be less correlated with the wind vectors and indicate a semi-diurnal signal.

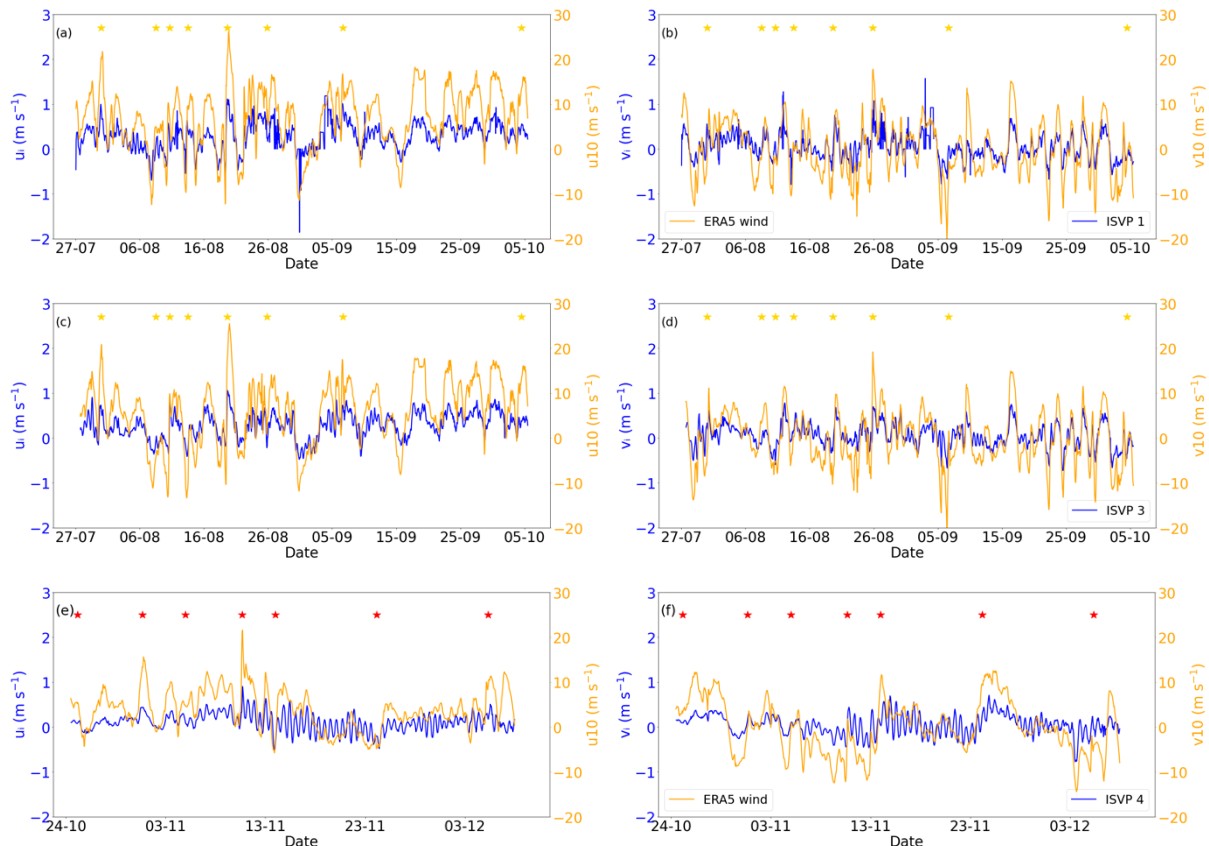

**Figure 5.** Velocity components of the buoys (on the left axis) and 10 m wind from the ERA5 reanalyses (on the right axis): the zonal and meridional component of ISVP 1 in winter **(a-b)**, of ISVP 3 in winter **(c-d)**, and of ISVP 4 in spring **(e-f)**. The star symbols are the same as in Fig. A1. Note that ERA5 wind magnitude is similar for the ISVP 1 and ISVP 3 locations.

The observed buoy speed and direction have been also compared with the ERA5 wind vectors to quantify the physical
control of atmospheric forcing on sea ice drift (Sect. 3.3). In order to remove the period of oceanic drift, ISVP 1 was only analysed for the first 10 days, when it plausibly drifted within both the SSMIS and AMSR2 ice edges. The main results of this analysis are summarised in Table 2. The Winter buoys exhibited high wind factors ranging between 3.16 % and 3.78 %, with small turning angles ranging between -7.89$^{\circ}$ and -11.19$^{\circ}$. The Spring buoys exhibited lower wind factors ranging between 2.42 % and 3.05 % and larger turning angles, between -21.18$^{\circ}$ to -27.00$^{\circ}$. This indicates that the drift of the Spring
buoys had a lower response to wind forcing but still in the range of the Antarctic values (Sect. 1) and high compared to the Arctic.

The relationship between the buoys' drift and the wind vectors can be quantified by both the vector $R_v^2$ and Pearson $R_p^2$ correlations (see Table 2). All p-values computed for the Pearson correlation were less than 0.05, thus indicating a
statistically significant correlation, and a linear relationship between the buoy and wind velocities.

**Table 2:** Parameters describing wind and ocean forcing on sea ice drift.

| Buoy | Mean wind factor (%) | Mean turning angle ($^o$) | Vector correlation $R_v^2$ | Pearson correlation $R_p^2$ | Mean current velocity (cm s$^{-1}$) | | Mean current speed (cm s$^{-1}$) |
|---|---|---|---|---|---|---|---|
| | | | | | $\overline{c_u}$ | $\overline{c_v}$ | |
| **Winter** | | | | | | | |
| ISVP 1 (first 10 days only) | 3.16 | -7.89 | 0.53 | 0.50 | 4.68 | 3.57 | 5.88 |
| ISVP 2 | 3.77 | -7.47 | 0.64 | 0.56 | 7.06 | 5.33 | 9.83 |
| ISVP 3 | 3.62 | -11.19 | 0.74 | 0.59 | 6.62 | 1.84 | 6.88 |
| **Spring** | | | | | | | |
| ISVP 4 | 3.03 | -21.18 | 0.54 | 0.27 | -0.73 | 0.73 | 1.04 |
| ISVP 5 | 2.89 | -23.59 | 0.56 | 0.35 | -1.33 | -1.07 | 1.71 |
| ISVP 6 | 3.05 | -21.87 | 0.54 | 0.37 | -3.12 | -0.01 | 3.12 |
| Trident* | 2.42 | -27.00 | 0.44 | 0.37 | 0.13 | -2.11 | 2.11 |

*Four-hourly time interval

The mean residual current velocity components ($\overline{c_u}$, $\overline{c_v}$) are estimated as described in Sect. 3.3 (Table 2). The mean current speed for the Winter buoys was larger than the Spring buoys, due to the Winter buoys drifting within the region of the ACC. Moreover, as the mean velocity components were positive in both the zonal and meridional directions, the underlying currents most likely also enhanced the wind-driven drift of the ice. On the other hand, the mean current velocity components of the Spring buoys varied greatly, and in most circumstances, they opposed the direction of wind-driven ice drift. This can be indicative of a region characterised by the presence of oceanic eddies. We note that while these residual currents are a rough estimate of the under-ice currents (see Appendix B), they do suggest that the ACC had a significant impact on the drift of the Winter buoys in comparison to the Spring buoys drifting further from the ice edge.

### 4.4 Winter-spring differences in the frequency domain

Figure 6 shows the spectra of the ERA5 wind and ice drift velocities of ISVP 1 (a), ISVP 3 (b), and ISVP 4 (c). The other buoys in each corresponding season indicated similar results to the ones displayed here and can be found in Figs. S9 and S10. The wind velocity spectra for both seasons form a typical, continuous energy cascade from the lower frequencies to the higher frequencies. In contrast with winter data from Womack et al. (2022), the drift of the Winter buoys revealed no apparent inertial oscillations (Fig. 6a and b), although a weak and statistically non-significant peak at 15.05 hours is detected. However, this is not large enough to make a comprehensive analysis of a possible shift to the inertial range. Also noteworthy is the "flattening" of ISVP 1's energy cascade in the high-frequency range (with a period smaller than 9 hours).

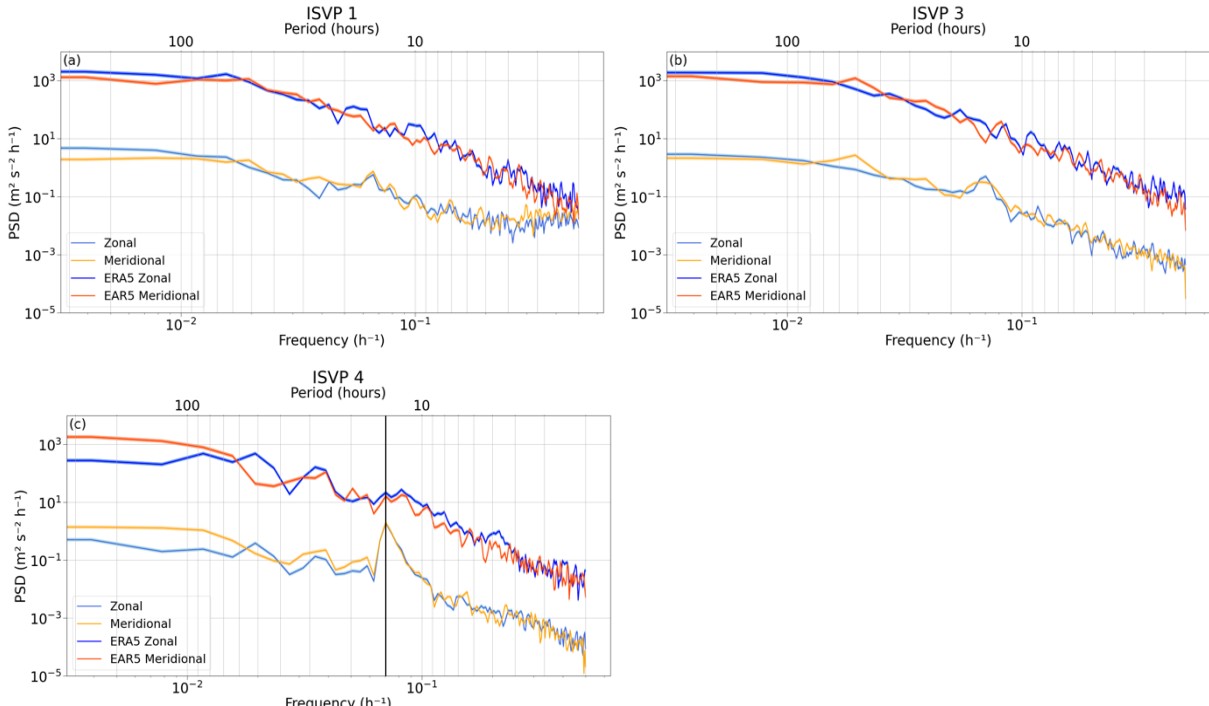

**Figure 6. (a)** Power spectral density corresponding to the zonal and meridional components of ISVP 1, in winter, and ERA5 wind velocity. **(b)** Same as in **(a)** but for ISVP 3, in winter. **(c)** Same as in **(a)** but for ISVP 4, in spring, where the black vertical line indicates the peak associated with inertial oscillations at 14.22 hours.

ISVP 4 and the rest of the Spring buoys exhibited an energy peak at the inertial frequency (highlighted by the vertical black line in Fig. 6c). The period of these oscillations is 14.22 hours for the ISVPs and slightly lower at 13.93 hours for the Trident, as it was deployed further south. In comparison, Heil et al. (2009) reported a period of 13.19 hours for the East Antarctic at ≈ 65° S. However, all Spring buoys indicated a similar inertial response within the theoretical inertial range, determined by the Coriolis parameter, at 57° S-60° S (14.30-13.85 hours). This can be clearly seen by the cyclic loops during their northward drift (Fig. 2b).

Figure 7 shows both the wavelet power spectrum (left panel) and the wavelet spectrum (right panel; this corresponds to the power spectrum integrated over time) of the filtered velocity magnitude for the reference buoys. The other buoys in each corresponding season indicated similar results and can be found in Figs. S11 and S12 of the supplementary material. Despite the use of the Butterworth high-pass filter, majority of the power (found within the cone of influence) in the Winter buoys' velocity spectra remained at the multi-day periods with peaks at 64, 128 and 256 hours (≈ 3, ≈ 5 and ≈ 10 days respectively). While the Winter buoys did respond to the cyclones differently, these intensifications found at the low frequencies can be associated to passing cyclones. This response was strongest during the fifth and sixth cyclones (19 and 25 August, respectively) when the Winter buoys switched from drifting north-eastwards to predominantly eastwards (Fig. 2a), under the action of winds with speeds > 25 m s$^{-1}$ (Fig. 5a-d). After the seventh cyclone (6 September), ISVP 3 continued to exhibit increased power at the low frequencies. We attribute this to the long period of high pressure between the 7-30 October (Fig. A1a), when strong winds persisted (Fig. 5a-d). ISVP 1 measured less power during this period, possibly because it drifted in between ice floes. Overall, these increased power intensifications at the lower frequencies were due to the direct transfer of momentum from the wind forcing. This can be seen in Fig. 6a and b where the drift spectra, although with less power, followed the energy cascade of the wind within the lower frequency range. This is in agreement with previous literature, where it was shown that the cascade of energy arises from non-linear ice dynamics (Heil and Hibler,

2002; Geiger and Drinkwater, 2005), and can eventually lead to inertial oscillations in the ice motion (Heil et al., 2009).
However, as indicated by the wavelet spectra (Fig. 7a and b; right panel), the Winter buoys (including ISVP 2 that drifted for half the time; Fig. S11) continued to exhibit no statistically significant power within the inertial range, even after the high-pass filter was applied. There were a few "pulses" of energy at the inertial frequency (Fig. 7a and b; left panel), but these were very short-lived and much weaker than the synoptic response of the ice cover. Furthermore, ISVP 1 continued to show increased and statistically significant power at the very high frequencies, which were not always associated with
passing cyclones.

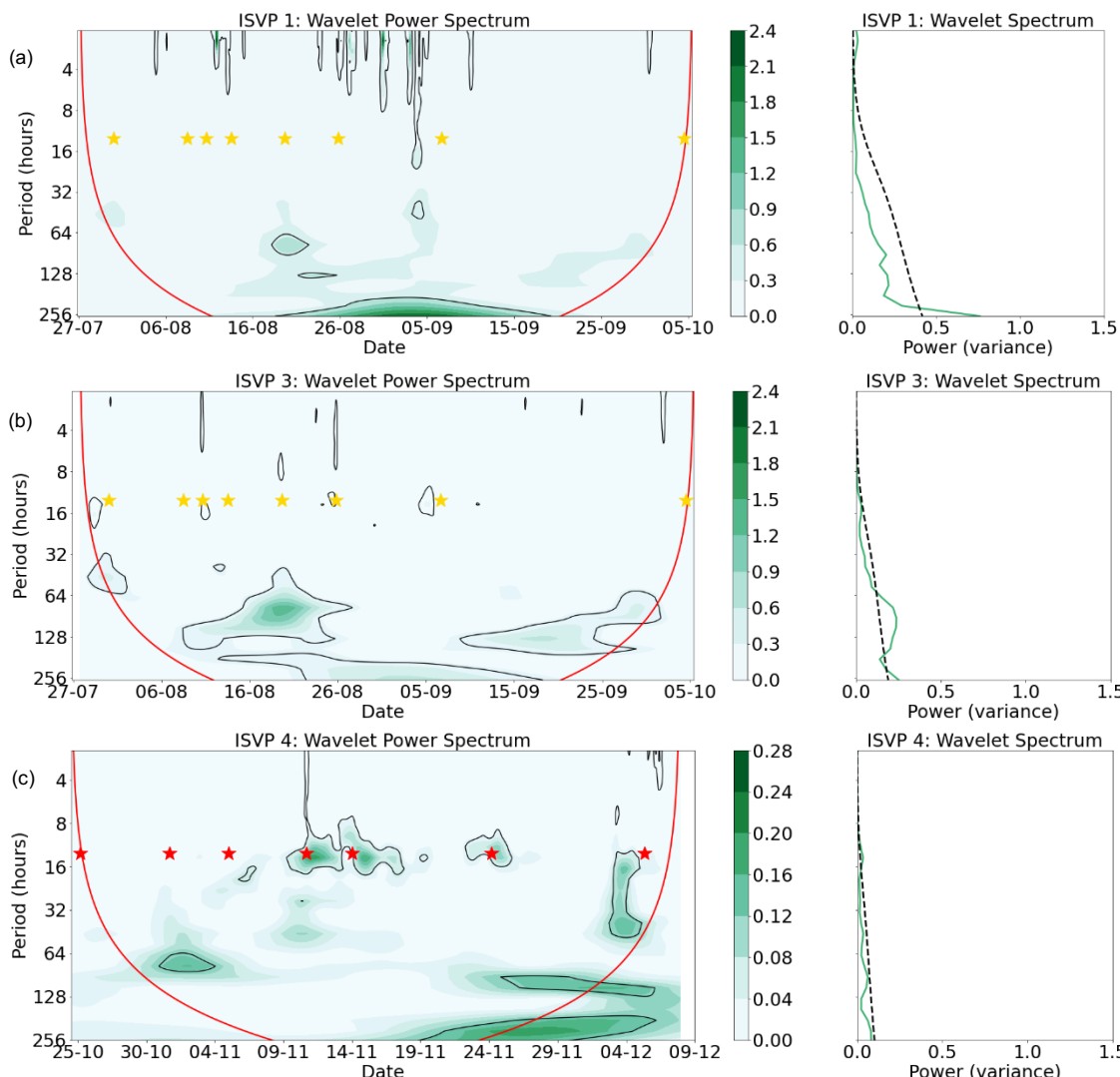

**Figure 7. (a)** The wavelet power spectrum with time (left panel) and wavelet spectrum (right panel) of the filtered velocity magnitude spectrum of ISVP 1, during winter. **(b)** Same as **(a)** but for ISVP 3, during winter. **(c)** Same as (a) but for ISVP 4, during spring. The red line indicates the cone of influence (left panel), the black contours (left panel) and black dashed lines (right panel) indicate the 95 % significance level. The star symbols are the same as in Fig. A1. Due to the difference in the intensity of the power between the seasons, the ranges are different between the Winter buoys and the Spring buoys.

The spring buoy, ISVP 4 (Fig. 7c), also indicated low-frequency power intensifications (within the cone of influence) that were associated with the passage of cyclones. However, compared to winter, the power of the Spring buoys at the multi-day periods was far weaker, and also statically non-significant in the wavelet spectrum (Fig. 7c; right panel). Rather, the majority of the power was found at the inertial frequency. This is because in addition to the synoptic response of the Spring buoys, the fourth to sixth cyclones (10-24 November) also generated strong inertial oscillations, which occupied a well-defined frequency band ($\approx$ 2 cycles day$^{-1}$). At a closer inspection, the dates of these strong inertial oscillations corresponded to the northwards drift of the Spring buoys when the ice drift velocity (Fig. 5e and f) was governed by high-frequency oscillations, with increased meandering and TSEs (Fig. 3b and f, respectively). These inertial oscillations continued until after the passage of the sixth cyclone, when they then dissipated within a few days, due to friction at the ice-ocean interface and/or the internal stresses caused from mechanical interactions within the ice (Colony and Thorndike, 1980; Leppäranta, 2011; Gimbert et al., 2012b; Lei et al., 2021; Marquart et al., 2021). The Spring buoys thus returned to a more straight-line drift path (Fig. 2b). However, another shorter-lived inertial response did occur before the passage of the seventh cyclone (5 December).

## 5 Discussion

### 5.1 Detecting ice type from drift features

We will first discuss the four Spring buoys, which were deployed during well-developed, first-year ice consolidated ice conditions (Fig. 1b), where the internal ice stresses are expected to be significant (Kawaguchi et al., 2019). Thus, while the sea ice drift followed the winds, the drift velocity exhibited a dampened signal (Fig. 5e and f). However, after the second cyclone (31 October), the air temperatures increased and began oscillating with a diurnal frequency near melting point (Fig. A1b and c). These sustained higher air temperatures, along with increased waves-in-ice activity during the passage of the second and third cyclones (31 October and 5 November; Thomson et al., 2023), most likely caused the consolidated ice cover to break up and melt. Subsequently, the ice cover became more susceptible to drift and deformation by the winds and ocean currents. This can be seen by the two earlier peaks in the daily meander coefficient time series (Fig. 3b), with the concurrent notable increases in the TSEs (Fig. 3f) and in the drift velocities (Fig. 5e and f). Since the ice cover would have become more susceptible to wind forcing, this led to the growth of the zonal component of the absolute dispersion (Fig. 3d) and hence, the initial 3°-5° eastwards displacement of the ice floes (Fig. 2b). Consequently, while the mean wind factors and mean turning angles indicated a dynamic response of the ice floes to wind forcing, we found little correlation of the overall drift velocity to the wind velocity, using the two different methods (Table 2). We later recomputed these two correlation parameters using ISVP 4 from its deployment date to 9 November (before the inertial oscillations were excited), and they both increased from $R_v^2 = 0.54$ and $R_p^2 = 0.27$ to $R_v^2 = 0.89$ and $R_p^2 = 0.78$, respectively. Therefore, although the power found within the lower frequencies of the wavelet power spectra (Fig. 7c; left panel) was non-significant, these measures suggest that during the first few days the spring sea ice was correlated to the wind forcing at the synoptic scale, as observed for the winter ice cover.

During the passage of the fourth to sixth cyclones (10-24 November), the associated transient winds allowed for the development of inertial oscillations within the drift (Fig. 5e and f). This is indicative of strong Coriolis forcing, which was likely larger than the advection term in the momentum balance. The inertial response influenced the spring sea ice at the shorter timescales and caused deviations of the ice drift from its initial more straight-line path, as seen by the large peaks in the daily meander coefficient (Fig. 3b), the erratically varying TSEs (Fig. 3f), and the change to the predominately

northwards drift of the ice floes (Fig. 2b). This led to the eventual dominance of the total absolute dispersion by the meridional displacement (Fig. 3d), and notable changes to the relative dispersion's meridional component (Fig. S6b). Additionally, the inertial response resulted in larger turning angles (Table 2), as Coriolis forcing notably deflected the ice floes left from dominant the wind forcing. The use of the filtered wavelet spectra highlights this strong response of the ice cover to the atmospheric forcing at the inertial frequency (Fig. 7c). This further indicates that the spring ice cover continued to break up into smaller floes and brash ice. Our results demonstrated that the spectra of both the drift and wind exhibited an energy cascade from the lower to higher frequencies; however, sea ice motion exhibited majority of its power at the inertial frequency, rather than at the same frequencies at which the atmospheric forcing was occurring (Fig. 6c). We attribute this to the weak geostrophic currents (Table 2), which were also observed in Geiger et al. (1998) and Alberello et al. (2020), and the increased mobility of the ice floes (Johnson et al., 2023). Furthermore, while our analysis indicates a plausible correlation between the presence of cyclones and the onset of the inertial oscillations, the power intensifications at the inertial frequency in some cases occurred outside the dates of higher cyclone activity (Fig 7c; Fig. S12 in the supplementary material). Following the analysis of Womack et al. (2022), we attribute this to the northwards drift of the sea ice while the ice edge was concurrently melting and retreating, where a likely increase in the propagation of waves may also have triggered the inertial oscillations or allowed the geostrophic current to keep the weaker oscillations during the periods of quiescence.

In winter, the mobility of sea ice is generally assumed to decay quickly due to its consolidation (Doble and Wadhams, 2006; Weeks and Ackley, 1986). However, the sampled sea ice continued to exhibit high drift velocities (Fig. 5a-d) and a dynamic response to wind forcing, i.e. the trajectories displayed periods of sharp turns and meanders (Fig. 2a), and periods of significant stretching and compression (Fig. 3e). This erratic nature of ice drift was particularly evident in the period between the second and fifth cyclones (8-19 August), as the daily meander coefficient and TSEs notably increased (Fig. 3a and e, respectively). This suggests that the ice drift was more tightly linked to the wind forcing during the passage of these three cyclones. In this case, contrasting to the 2017 buoy analysed by Womack et al. (2022), the Winter buoys were forced closer to the ice-edge region, and the mean turning angles (Table 2) were notably smaller than the value of -19.83$^{\circ}$ reported by Womack et al. (2022) for pancake ice, and the values presented by the Spring buoys, drifting further from the ice edge. This indicates that the ice floes near the ice edge drifted more closely to the direction of the winds. Moreover, the mean wind factors (Table 2) exhibited higher values than 2.73 % by Womack et al. (2022). They were instead closer to the median wind factor of 3.9 % reported by Wilkinson and Wadhams (2003) for low sea ice concentrations ($\leq$ 25 %). Since the wind factor and turning angle are known to be modified by the underlying ocean current (Nakayama et al., 2012), and in conjunction with the final cumulative meander coefficient of < 1.5, we suggest that the high velocity of the ACC (Table 2) provided a steady source of significant energy. This in turn modulated the relationship between the buoys' drift velocity and the wind vectors. Subsequently, this led to the enhancement of the wind-driven ice drift by the eastwards flowing ACC, which allowed ISVP 1 and ISVP 3 to travel over 25$^{\circ}$ (> 1500 km) eastwards in 70 days (Fig. 2a; Table 1). Together, the analysis indicates that the 2019 winter sea ice was under a much stronger steering influence of the ACC than the ice floe analysed by Womack et al. (2022), which drifted > 200 km from the sea ice edge and further away from the more intense region of the ACC.

The spectral analysis provides an additional argument. While the wind- and ice-drift velocity components exhibited the typical continuous energy cascade from the lower frequencies to the higher frequencies (Fig. 6a and b), this energy cascade did not generate any statistically significant peaks within the inertial range (Fig. 7a and b). Majority of the power instead continued to be found at the same frequencies at which the large-scale atmospheric forcing occurred. The amount of power found within these lower frequencies of the wavelet power spectra was also considerably larger than in Womack et al.

(2022; their Fig. 13). Geiger et al. (1998), in the western Weddell Sea, also found that the Antarctic Coastal Current provided a steady source of moderate low-frequency power to the ice drift. Therefore, the results further confirm that the strong wind and oceanic forcing together caused the nonlinear velocity terms to remain much larger than the Coriolis term. Consequently, the winter sea ice did not exhibit a clear energy peak within the inertial range. Together with the significant correlation values (Table 2), this high mobility of the winter ice cover suggests that the heterogeneous ice conditions like the ones observed during deployment (Fig. 1a) were maintained, and hence free-drift conditions are likely to persist throughout winter in this region. However, the evidence of no inertial oscillations is in stark contrast to the data reported by Womack et al. (2022) for free-drift conditions. We therefore confirm the stronger role of the underlying current in the region of this study.

We now briefly discuss the features of the Winter buoy, ISVP 1, which was deployed in the interstitial ice between ice floes. Its diagnostics indicated a significantly lower correlation to the winds of only $R_v^2 = 0.53$ and $R_p^2 = 0.50$ (Table 2), even though these parameters were computed for the first 10 days when it drifted within both the SSMIS and AMSR2 ice edges (Fig. S1 in the supplementary material). It also exhibited an elevated higher frequency portion in its drift spectra (Fig. 6a). This phenomenon was also observed by Doble and Wadhams (2006) for an outer ice-edge buoy, during the pre-consolidation phase. When we recomputed the power spectrum for the first 10 days, the elevated higher frequency portion was reduced (Fig. S13 in the supplementary material). This is indicative of increased oceanic forcing in the signal. Moreover, the dates of this higher-frequency signal in the wavelet power spectrum (Fig. 7a) additionally corresponded to the dates of the extreme peaks ($> 1$ m s$^{-1}$) in ISVP 1's drift velocity (Fig. 5a and b), and peaks in its daily meander coefficient and TSE (Fig. 3a and e, respectively). Therefore, we can confirm that ISVP 1 left the ice cover during these periods. In this regard, we warn on the limitations of determining ice type conditions from floating drifters, since they would confound the relationship to wind forcing and eventually the deformation rate.

## 5.2 Seasonal and regional comparison of the deformation proxy

The differential drift of sea ice in each season was principally examined through the evolution of the deformation proxy $\sigma_{\dot{D}}$ (Fig. 4), in relation to the passage of cyclones. During spring, although the buoy clusters indicated coherence in its total relative dispersion (Fig. S6b), the associated spring $\sigma_{\dot{D}}$ exhibited large variations that coincided with the occurrences of the cyclones (Fig. 4b). Whilst this proxy cannot discern whether this is due to divergence, convergence and/or shear, it is likely due to the varying meridional winds which resulted in the changes in the meridional separation, and more erratic changes in the TSEs (Fig. 3f). The $\sigma_{\dot{D}}$ was additionally characterised by regular higher-frequency changes. A spectral analysis of the $\sigma_{\dot{D}}$ (Fig. 8b) confirmed these changes to be at a period of 13.57 hours, which is situated within the theoretical inertial range. In contrast to the analysis of the Spring buoys' drift velocities themselves, the spectra exhibited no energy cascade from the lower to the higher frequencies. Power associated at the lower frequencies, prevalent for the sea ice velocity, was effectively dampened out. The lower frequencies were strongest at $\approx 5$ days, and associated with the occurrences of cyclones, with secondary peaks at the lower and inertial frequencies. Similarly, Heil et al. (2008) found that, for buoys further from the Antarctic coastline, the power spectrum was dominated by a peak at the diurnal frequencies with secondary peaks occurring at the inertial and lower frequencies. On the other hand, in coastal regions around Antarctica, the power in sea ice deformation is rather driven by sub-daily processes without any low-frequency contributions for all kinematic parameters (e.g. Geiger et al., 1998; Heil et al., 2008, 2009, 2011). This difference is attributed to the different bathymetry, where buoys drifting near the coast are more susceptible to tidal forcing (Heil et al., 2008), which can enhance the sub-daily signal. Collectively, our results indicate a counterintuitive strong de-coupling

between the large-scale atmospheric forcing and the sea ice deformation in spring, despite the expectation that fractured sea ice would be prone to following wind forcing. In the western Weddell Sea, Geiger et al. (1998) showed that while moderate low-frequency currents must also have an effect, the sub-daily and daily deformation processes were driven by the wind-induced inertial oscillations of the ice-ocean system. They additionally demonstrated that that spatial features in the underlying current, due to topological features, showed in the non-linear interactions. Thus, while bottom topography would not have had any effect on the deformation of spring sea ice in this open-ocean region, Swart et al. (2020) speculated that following the ice melt, the interactions of freshwater input and intense winds of the Southern Ocean can promote and alter sub-mesoscale eddies. This may be an energy source contributing to the shape of the spring deformation spectra at the higher frequencies (Fig. 8b) in addition to the high-frequency ocean oscillations, while lower frequencies were more likely coupled to the intermittent winds and the weaker geostrophic currents.

The winter deformation proxy (Fig. 4a), along with its corresponding dispersion analyses (Figs. 3c and S6a), showed pronounce fluctuations, throughout the analysed period, that were notably related to the occurrences of cyclones. This was due to the high mobility and dynamic response of the winter ice floes to the winds and ACC. Therefore, the winter ice cover was unable to transmit the stress necessary to resist deformation by the passage of the cyclones, and the associated wind directional changes (Fig. 5a-d). In agreement with Hutchings et al. (2011), we find that since the internal ice stresses were not significant in winter, the sea ice velocity (Fig. 5a-d) and deformation time series followed the wind forcing on the ice cover, and thus remained linearly related to the large-scale atmosphere. This was further exemplified by the power spectrum of the winter $\sigma_{\dot{D}}$ (Fig. 8a). The $\sigma_{\dot{D}}$ spectra similar to the velocity spectra formed a typical, continuous energy cascade from the lower to higher frequencies, with majority of the power found at the same frequencies at which the large-scale atmospheric forcing was occurring (Fig. 6a-b). In accordance with Geiger et al. (1998), we suggest this was additionally due to the steady source of low-frequency energy from the ACC, which was in strong coherence with the wind. Therefore, unlike the spring $\sigma_{\dot{D}}$ and previous literature, the winter $\sigma_{\dot{D}}$ like its drift velocity remained strongly coupled with the atmosphere in winter.

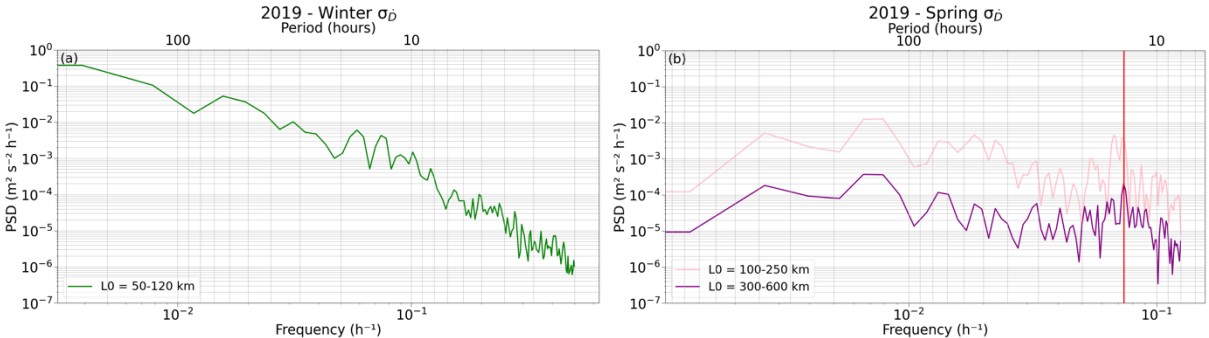

**Figure 8. (a)** Power spectral density of the winter deformation proxy $\sigma_{\dot{D}}$. **(b)** Same as in **(a)** but for spring, where the red vertical line indicates the peak associated with inertial oscillations at 13.57 hours). L0 denotes the horizontal spatial scale for each cluster of buoys. The line colours correspond to the $\sigma_{\dot{D}}$ time series in Fig. 4a and b.

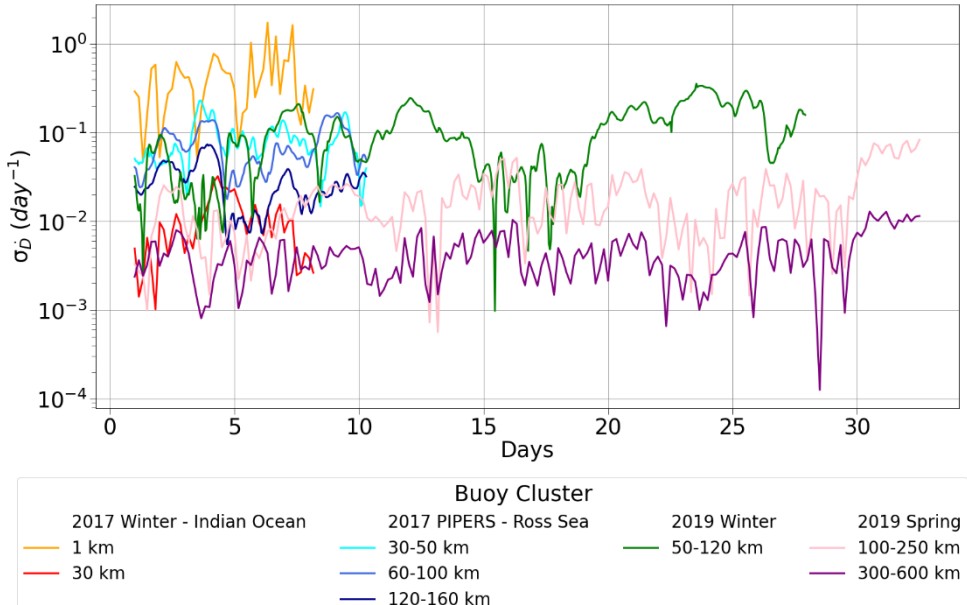

**Figure 9.** The standard deviation $\sigma_{\dot{D}}$ of the deformation rate $\dot{D}$ versus the time interval, in days from start of analysis, for buoys from the 2017 Winter Cruise, 2017 PIPERS winter campaign, and the 2019 Winter and Spring Cruises. Each cluster of buoys has been denoted below, with their corresponding initial horizontal spatial scales (L0).

The sea ice deformation is scale dependent as the ice velocity field is known to be spatially discontinuous (Lindsay et al., 2003). The two horizontal spatial scales of the spring cluster allowed for a comparison of the deformation across different spatial scales. Our analysis shows that as the spatial scale decreases, larger strain rates become apparent, which we find in
agreement with previous literature (e.g. Rampal et al., 2008; Hutchings et al., 2011; Lindsay et al., 2003). This indicates that the largest strain rates (and gradients) are held by the smaller portion of the sea ice area. Another signature of this deformation localisation at smaller spatial scales, is the increase of $\sigma_{\dot{D}}$ power as the scale is reduced, which is in agreement with Marsan et al. (2004). However, very few buoys have been deployed in the Antarctic MIZ to allow for a full comparison of the deformation state under different seasonal conditions or across regions. We provide an initial summary by including
additional data available to us from two buoy arrays deployed in different regions around Antarctica. The first was deployed during the 2017 Winter Cruise at approximately 62$^{\text{o}}$ S and 30$^{\text{o}}$ E (Machutchon et al., 2019; Vichi et al., 2019; Alberello et al., 2020; Womack et al., 2022), and are analysed in this study between 5-13 July 2017. The second array included six of the fourteen buoys from the 2017 Polynyas, Ice Production, and seasonal Evolution in the Ross Sea expedition (PIPERS) winter campaign, which were deployed at approximately 67$^{\text{o}}$ S and 180$^{\text{o}}$ E (Kohout et al., 2020). These six buoys are
analysed in this study between 13-23 June 2017.

The results from Fig. 9 show an overall reduction of deformation as the area, over which it is computed, is increased. However, the 2017 Winter Cruise buoys, and more noticeably at the 30 km spatial scale (red line), exhibited a lower magnitude than the 2017 PIPERS and the 2019 Winter buoys of similar spatial scales. While all three of these buoy arrays
were deployed on pancake-ice conditions (Alberello et al., 2020; Kohout et al., 2020), where we would assume a similar rheology of the ice cover, the $\sigma_{\dot{D}}$ of the 30 km cluster was rather more similar to that of the 2019 Spring buoys, but with significantly larger spatial scales. As waves help to maintain the pancake-frazil ice conditions (Weiss and Marsan, 2004; Kohout et al., 2014; Vichi et al., 2019), we attribute this to the varying waves-in-ice conditions occurring during the drift of the three winter-buoy arrays. Alberello et al. (2020, their Fig. 3) reported significant wave heights of 0-6.25 m measured
by one of the 2017 Winter buoys, and drift speeds which predominantly fluctuated between 0-0.5 m s$^{-1}$ in mostly 100 % ice concentration. Kohout et al. (2020; their Fig. 4a) reported larger maximum significant wave heights of $\approx$ 2-9 m and

drift speeds reaching $\approx 1$ m s$^{-1}$, between 13-23 June 2017, in a region of > 80% ice concentration. Therefore, as the wave heights and drift speeds were generally lower during the 2017 Winter-buoy drift, their freedom to respond to wind and ocean forcing most likely was reduced, possibly due to the close packing of the floes. This would have resulted in a more significant ice rheology and thus, a higher resistance to deformation by the winds and ocean currents. Therefore, while we can confirm that the estimated deformation is a function of the area over which it is calculated, we find that in the Antarctic it is also strongly determined by the rheology of the ice cover, which can be preconditioned from previous seasons (Lei et al., 2020), as well as the atmospheric and oceanic forcing under which it is governed. Consequently, Antarctic sea ice may not always exhibit a classic de-correlation length scale, which is in agreement with Hutchings et al. (2011) for the Arctic. Thus, the magnitude of deformation, for similar spatial scales, may vary between different seasons, regions and proximity to the sea ice edge and the Antarctic coastline.

**6 Conclusions**

In this study, we analysed a unique dataset of drifting sea ice buoys in the Atlantic MIZ in winter and spring 2019, which showed how the evolution and spatial pattern of sea ice drift and deformation were affected by the balance between external atmospheric and oceanic forcing and local ice conditions. We compared the estimates of deformation with the few datasets from other Antarctic regions and seasons. Here, we highlight our results and conclusions:

- During winter, the ACC modulated the relationship between ice drift and wind forcing, near the sea ice edge. This led to a highly energetic and mobile ice cover that was characterised by free-drift conditions, predominantly in the zonal direction. The resulting drift and deformation were primarily driven by large-scale atmospheric forcing, with a negligible inertial response. The relationship between wind forcing and sea ice drift and deformation both remained linear in winter, with the kinematic parameters strongly coupled to the atmospheric forcing.
- During spring, sea ice drift was initially driven by large-scale atmospheric forcing, but with a dampened signal due to the consolidated ice conditions at deployment and stronger internal stresses. As the surface heat balance changed, it caused the ice cover to melt and break up. This led to an increase in the drift kinematics, and the ice drift changed to be dominated by the inertial response. However, the deformation spectra in spring indicated a strong de-coupling to large-scale atmospheric forcing, which is counterintuitive in melting conditions. We interpret it as an increased response to the ocean forcing, with higher frequencies driven by ocean oscillations and possibly influenced by sub-mesoscale flows.
- A comparative analysis of the existing datasets revealed that Antarctic sea ice deformation is strongly determined by the rheology of the sea ice, as well as the atmospheric and oceanic forcing on the ice cover. This indicates that a classic decorrelation length scale may not always exist. Therefore, the magnitude of deformation may vary with ice type between seasons, regions and the proximity to the ice edge and the coastline, for similar spatial scales.

In summary, the present study highlights the need for a better understanding of the impacts of ocean currents and waves on the Antarctic marginal ice zone to help fully understand and quantify the effects of atmospheric forcing, which is one of the main drivers of drift and deformation. The paucity of oceanic observations, especially in the ice-covered Southern Ocean has meant that these drivers are poorly constrained. Therefore, the collection of oceanic in situ observations is vital. These should include (1) measurements surface currents beneath the ice so that we can discern ice drift and deformation due to wind or ocean currents, and (2) the continuation of waves-in-ice measurements to not only understand how waves maintain a mobile ice cover, but to also understand how sea ice attenuates the wave energy. Additionally, buoys that are

ice-tethered and sink when ice melts, acting as an effective Lagrangian tracker of the ice itself, should become a priority, although environmentally undesirable. They avoid the aliasing of drift information and a more reliable detection of the ice floe lifetime, without making assumptions based on the remotely sensed edge detection. This is particularly important as there are known errors in satellite products, especially at the ice edge and during the melt season.

**Appendix A: Spatial and temporal changes in atmospheric conditions**

The time series of the ERA5 mslp and 2 m air temperature in the vicinity of the Winter and Spring buoys are presented in Fig. A1. We only present data from ISVP 1 since all three Winter buoys remained close together (Fig. 2a) and their meteorological conditions were very similar. Subsequently, ISVP 6 was also used as a reference for the eastwards cluster of the Spring buoys (Fig. 2b), and ISVP 4, that drifted $> 5^{o}$ ($> 400$ km) west from the rest of the Spring buoys and under slightly different meteorological conditions, has been included as well. The time series for all buoys can be found in the supplementary material (Figs. S3 and S4).

During the passage of each cyclone, highlighted by the stars, there was a characteristic drop in the mslp, in both winter and spring. However, there was a significant difference in the time series of the air temperatures between the two seasons. During the winter deployment, the air temperatures initially fluctuated between $-15^{o}$C and $0^{o}$C as warm air was advected poleward on the eastern flank of cyclones, while the cold polar air was advected equatorward on the western flank (Schlosser et al., 2018; Vichi et al., 2019). However, as the Winter buoys drifted into September and October, this fluctuation became smaller, along with the gradual increase in the background atmospheric temperature. The air temperatures overlying the region of the Spring buoys exhibited a similar fluctuation pattern compared to the end of the wintertime series for the first five days, as shown by ISVP 4 (Fig. 2b). After the second cyclone (31 October), the air temperatures fluctuated daily between $-5^{o}$C and $0^{o}$C. Although temperatures increased slightly during the passage of the cyclones, the daily signal appeared to be dominant in contrast to the winter conditions.

All the analysed cyclones (minimum core pressures $< 970$ hPa) carried substantial energy in their winds, with speeds $> 16$ m s$^{-1}$ in winter and $> 10$ m s$^{-1}$ in spring. Cyclones are also prominent transporters of moisture and heat to the polar latitudes (Messori et al., 2017), and generate large waves in the Southern Ocean that can propagate hundreds of kilometres into the ice cover (Kohout et al., 2014). The impacts of these cyclones on the ice cover were however different during each season, as detailed in the succeeding sections.

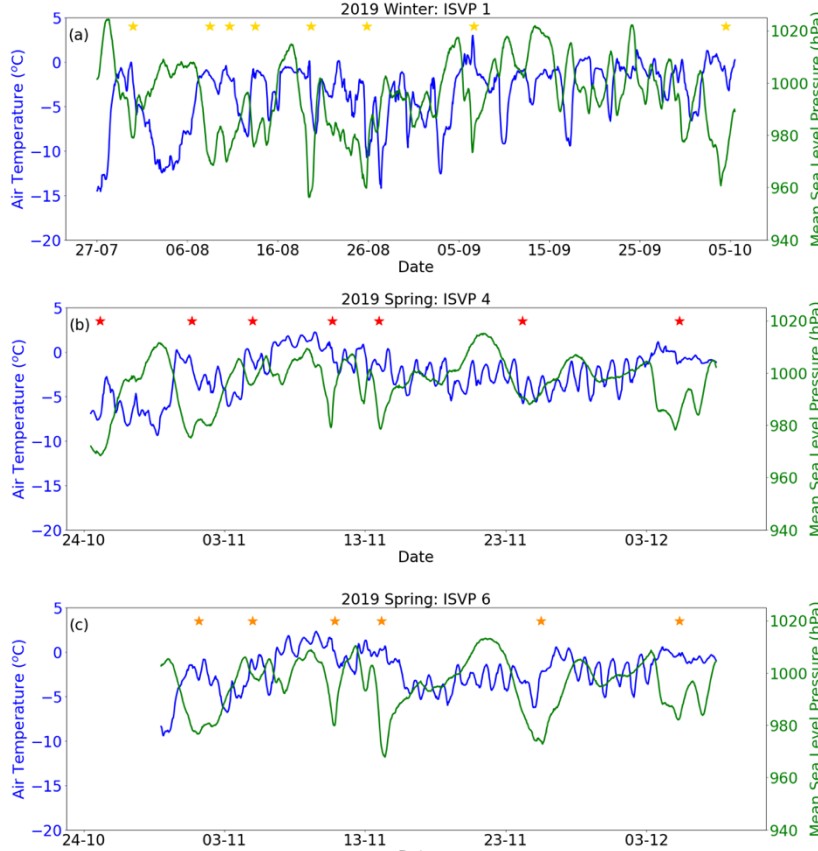

**Figure A1.** Time series of the ERA5 2 m air temperature (blue) and mean sea level pressure (green) at the location of ISVP 1 in winter **(a)**, IVSP 4 **(b)** and ISVP 6 **(c)** in spring. The gold stars denote when the winter cyclones were closest to the buoys. The red stars denote when the spring cyclones were closest to ISVP 4. The orange stars denote when the same spring cyclones were closest to the main spring cluster deployed > 5° (> 400 km) east of ISVP 4.

**Appendix B: Comparison of residual currents and the Copernicus GlobCurrent database**

While there are no in situ under-ice currents available to validate the residual currents, there is a remoted sensed product that can be used to compare the estimate of the winter residual currents, and check if they are comparable with the velocities

observed in the ACC outside of the sea ice at this longitude. Herein, we use data of current velocity from Copernicus GlobCurrent database in the region of the Winter buoys' trajectories. This dataset integrates the velocity field of geostrophic surface currents from satellite products recorded from 1993 to 2019 (Rio et al., 2014) and modelled Ekman currents, which contain components from wind stress forcing provided from atmospheric and drifter data (Derkani et al, 2021).

Fig. B1 (left-hand-side) shows the mean current speed for the period of the Winter buoys' deployment (27 July 2019 to 5 October 2019). The GlobCurrent database does not have any data for the latitudinal range of the Spring buoys, since they were deployed and drifted further from the sea ice edge. We reiterate that this product is a model, and as reported in Section 2, there is are still significant biases for models in the Southern Ocean. Therefore, the GlobCurrent database does have uncertainties (Derkani et al., 2021), including the fact that geostrophic currents cannot be computed from altimetry under

the sea ice. It is more likely that the velocities are biases towards the open ocean components far from the ice edge. The mean zonal and meridional velocities, for the duration and between -55° S to -56° S (the mean of the buoys' latitudes) and 0-26° E (as indicated by the cyan rectangular box), were extracted to compute the distribution of the velocity components (right-hand-side). The magnitude of the winter residual currents is additionally shown, where it can be seen that they fall within the range of the GlobCurrent distribution, although in the lower tail. This is compatible with the under-ice

 momentum reduction of ocean currents. The aim was to however demonstrate that the latitude of the 2019 winter deployment is more likely to find more intense ocean currents than at the latitudes of the 2017 winter and 2019 spring deployments. This indicates that currents can be expected to be even higher during the 2019 winter deployment, which supports our hypothesis.

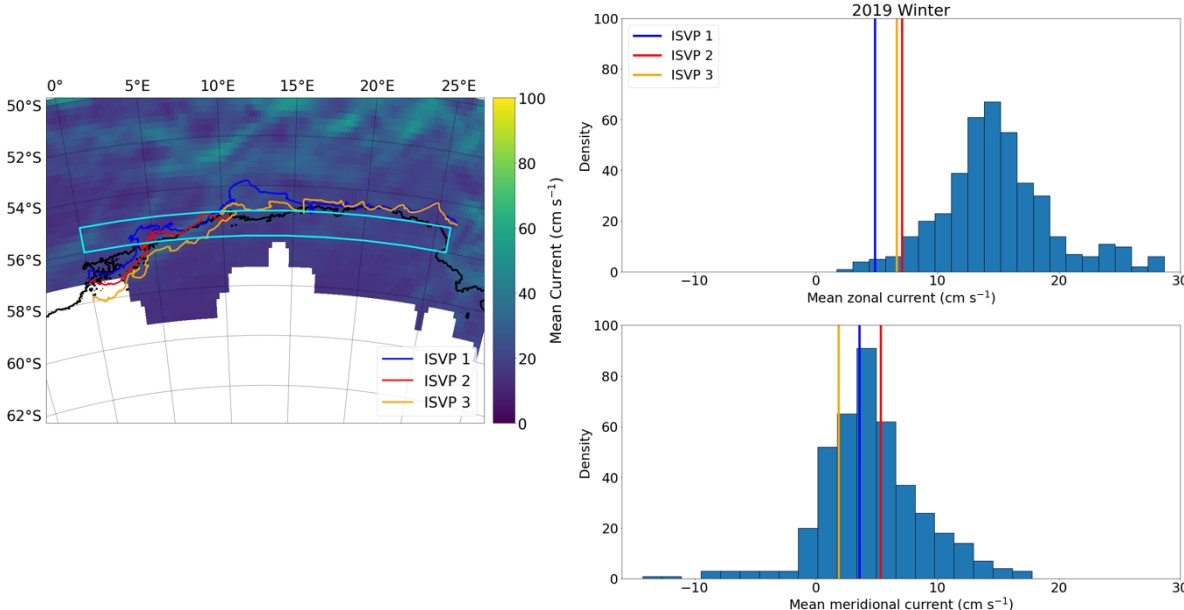

**Figure B1.** (left-hand side) The mean Copernicus GlobCurrent speed for the deployment period of the Winter buoys (27 July 2019 to 5 October 2019). The trajectories of the three Winter buoys are shown. The black contour denotes the AMSR2 0 % sea ice concentration on 30 September 2019 – the date of approximate austral sea ice maximum. The cyan rectangular box indicates the region used to compute the distribution of the zonal and meridional currents. (right-hand side) The distribution of the spatial mean of the GlobCurrent zonal and meridional currents. The vertical lines denote the magnitudes of the winter residual currents.

**Code and data availability:**

This study makes use of various data sets with different availability. The ERA5 reanalysis product at single-levels is available at https://cds.climate.copernicus.eu/cdsapp#!/dataset/reanalysis-era5-single-levels?tab=overview (Copernicus Climate Change Service (C3S), 2017). The GlobCurrent database is available at https://data.marine.copernicus.eu/product/MULTIOBS_GLO_PHY_REP_015_004/description (Copernicus Marine Service Information (CMEMS), 2014). The sea ice concentration data was obtained from the passive microwave Advanced Microwave Scanning Radiometer 2 (AMSR2) sensor (Spreen et al., 2008) at ftp://ftp-projects.cen.uni-hamburg.de/seaice/AMS2/, and the Special Sensor Microwave Imager/Sounder (SSMIS) product (NSIDC, 2023) at https://data.marine.copernicus.eu/products. The latter is no longer available there, and may be accessed from the Copernicus Climate Change Service at https://cds.climate.copernicus.eu/cdsapp#!/dataset/satellite-sea ice-concentration?tab=overview (Copernicus Climate Change Service (C3S), 2017). The in situ ISVP data is available at https://doi.org/10.5281/zenodo.7954779. The in situ Trident data is available at https://doi.org/10.5281/zenodo.7954841. The code used to process the data and produce the figures is available at https://github.com/mvichi/antarctic-buoys/.

**Supplement:**

The supplement related to this article is available online at:

**Author contribution:**

A.W. conducted all analyses and wrote the original draft with editing reviews from A.A, M.d.V., A.T. and M.V. The in situ buoy data were supplied by M.d.V. (ISVPs) through the South African Weather Service and R.V. (Trident) through the Department of Electrical and Electronic Engineering. M.V., A. T., and M.d.V procured the financial support.

**Competing interests:**

The authors declare that they have no conflict of interest.

**Acknowledgements:**

This expedition has been supported by the National Research Foundation of South Africa (grant no. 118745) through the South African National Antarctic Program. This work has received funding from the European Union's Horizon 2020 research and innovation programme under grant agreement no. 101003826 via project CRiceS (Climate Relevant interactions and feedbacks: the key role of sea ice and Snow in the polar and global climate system). A.T. acknowledges the support from the Australian Research Council (DP200102828). A.A acknowledges the support from the London Mathematical Society (Scheme 5 – Ref. 52206). The authors would like to thank the South African Weather Service (SAWS) for the usage of the ISVP data. We acknowledge the *Southern oCean seAsonaL Experiment* (SCALE), and thank the captain and the crew of the SA Agulhas II for the assistance during the deployments.

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
