# Peer review of "A contrast in sea ice drift and deformation between winter and spring of 2019 in the Antarctic marginal ice zone"

_EGUsphere, 2023_

## Author Response (AR1)

*Reviewer 1 – Ruibo Lei:*

*We would like to thank the reviewer for their time and effort taken to read and provide helpful comments and advice on our manuscript as he has aided in the analyses of this work. We have responded to all comments below.*

*We would like to first note that we have strengthened the motivation and conclusions for this study, and also streamlined the text. The old Figure 2 (atmospheric variables) and its explanation has now been moved to Appendix A (page 26-27), and the relative dispersion figures have been moved to the supplementary material (Figure S6). In this document, the page numbers, line numbers, and figure numbers that we refer to in our answers relate to the revised manuscript (marked up version). All answers have been italicised.*

Main comment:

My main concern is that if you cannot determine whether the buoy is on ice floe, how do you know if the motion characteristics are sea ice or current? Can you segment the data segments through analysis of motion characteristics to limit the observation data on ice. The study region south of the sea ice edge cannot be used to determine whether the buoy is on the ice.

*Answer: We thank the reviewer for this suggestion. We are confident that we have adequately identified the period when the uncertain buoys were drifting on ice, and we have ensured that this is clearly stated in the revised manuscript.  Since commercial ISVPs can continue to drift after the ice has melted or even if they exited the ice edge, there was uncertainty on whether the Winter buoys remained within the MIZ during their drift. This was particularly true for ISVP 1, as it was deployed in between pancake ice as reported in Section 2, line 153 of the revised manuscript. Therefore, we used two satellite remote sensing products to determine the distance of the buoys from the ice edge, and hence, the most likely dates when the buoys left the ice cover, which is a technique that is commonly applied (e.g. de Vos et al., 2021; Womack et al., 2022; Wahlgren et al., 2023; Nose et al., 2023). ISVP 1 remained within both the AMSR2 and SSMIS ice covers for the first 10 days (lines 209-211). This has now also been included in Table 1, page 5. Subsequently, our analysis of the wind response for ISVP 1 was only computed for those initial 10 days (see Table 2, page 17). We additionally recomputed the power spectral density of ISVP 1 for those initial 10 days (Figure S13 of the supplementary material), where in the discussion we report that the elevated higher-frequency portion of the power spectrum was reduced (see page 20, lines 780-781 of revised manuscript). This higher frequency elevation was attributed to oceanic forcing. Moreover, as reported in lines 782-784, the dates of the higher-frequency signal in the wavelet power spectrum (Figure 7) additionally corresponded to the dates of the extreme peaks ($> 1$ m s$^{-1}$) in ISVP 1's drift velocity (Figure 5a and b), and in its daily meander coefficient (Figure 3a), and the newly computed trajectory stretching exponents (TSEs; Fig. 3e). Therefore, we confirm that ISVP 1 did leave the ice cover during those periods, which additionally lead to our conclusion that ice-tethered buoys are more suitable for studying sea ice dynamics in the Antarctic (lines 784-786 and lines 922-925), and that floating buoys may lead to misattributions of sea ice drift features.*

*We have made sure that there is no ambiguity in the wording of the revised manuscript, nevertheless we believe that there is no need to add further evidence for ISVP 1 beyond what already included in this version of the manuscript. Our analysis of the drift kinematics for the other two Winter buoys, ISVP 2 and ISVP 3, indicated that they remained well within the ice cover (for the analysis period) as their drift velocities did not indicate extreme magnitudes, their PSD and wavelet analyses did not indicate an elevated high-frequency portion, and lastly their wind response was greater as the ACC had less steering influence on their drift.*

**Page 1, Line 20:** "For this highly advective coupled ice-ocean system, ice drift and deformation linearly depends on atmospheric forcing": Even for the highly advective coupled ice-ocean system, it still cannot measure the ice drift and deformation linearly depends on atmospheric forcing.

*Answer: We acknowledge this comment by the reviewer, which is indeed legitimate given the highly non-linear nature of air-ocean interactions. However, as winds transfer momentum to the sea ice cover, and under free-drift*

*conditions, the sea ice changes from moving to the left of isobars (Southern Hemisphere) to moving almost parallel to the isobars, with a linear relation to the surface wind velocity (Kottmeier et al., 1992; Wassermann et al., 2006). Therefore, since (i) no statistically significant inertial oscillations were exhibited within the Winter buoys' velocities (Figures 5, 6 and 7), (ii) the mean underlying currents enhanced the wind-driven drift, and (iii) free-drift conditions persisted throughout the deployment period with high linear correlation with the wind, the Winter buoys drifted closely with the overlying atmospheric forcing. This is highlighted by both the drift and deformation spectra which exhibited the typical energy cascade (Figures 6 and 8), similar to that of the ERA5 wind spectra. For this reason, we report that they were linearly related to atmospheric forcing. In the revised manuscript, we have highlighted the linear relation to the wind factor in the introduction (line 58), as well in the discussion 842-844).*

**Page 1, Line 44** "During the spring retreat season, the surface radiative balance changes, causing the consolidated ice to break up" suggest changing to "surface heat balance changes and enhance ice dynamics", it is not just radiative balance.

*Answer: Thank you for this suggestion. It will be replaced for all.*

**Page 4, Line 117:** ISVP -- the "i" here is the ice, not Iridium.

*Answer: We thank the reviewer for this relevant comment, however in this case for commercial surface velocity profilers, "i" does stand for Iridium (as opposed to Argos) and the leading "p" is for polar when they are optimised for ice tracking (i.e. p-iSVP). We would thus prefer to keep the original text.*

**Page 6, Line 190:** the positioning is through the GPS system, but the Iridium  The positioning accuracy of Iridium is very low.

*Answer: We acknowledge this suggestion. The position is derived by the buoys' GPS, and then it is communicated via the Iridium system. This has been corrected in the revised manuscript, line 242.*

**Page 7, Line 215:** what is the meaning of "(0.04 da"

*Answer: This was mistake – was not fully removed. It was the frequency for the high-pass filter (0.04 day$^{-1}$).  It will be removed in the revised manuscript.*

**Page 7, Line 217:** I don't know what is the meaning of "these buoys were near a tidal node".

*Answer: A tidal node is another term for an amphidromic point. This is a geographical location of the ocean where the tidal fluctuation (or tidal amplitude) is notably small (between 20-60 cm at most). Due to the ambiguity, we have clarified this in the revised manuscript, lines 271-272.*

**Page 11, Figure 3:** a table is need to further describe the sea ice conditions at each buoy deployment site. Ice concentration along the trajectory of each buoy also is needed to show.

*Answer: We acknowledge this suggestion by the reviewer. Figure 1 shows photographs of the ice conditions during the deployments of the buoys in each season. The pancake ice conditions occurred for all three Winter buoys and the consolidated ice conditions occurred for all four Spring deployments. The features of the sampling sites have been previously detailed in Skatulla et al. (2022) and Johnson et al. (2023). We acknowledge the importance of adding more detail in the manuscript, and hence we have included the citations above and additional information on floe size and thickness during each expedition in lines 151-152 and 166-167 of the revised manuscript.*

*We are reluctant to present SIC data along the trajectory, although we are aware that they have been used to determine the atmospheric reanalyses fields. These data may lead to misleading conclusions on the ice features. We recently demonstrated that remotely sensed SIC in the MIZ is not a good estimator of the ice type (Alberello et al., 2019; Vichi, 2022). In the supplementary material (Figures S1 and S2), we also highlight how different the two satellite remote sensing products (AMSR2 and SSMIS) can be in detecting the sea ice edge. Very little field data, for the validation of metocean (meteorological and oceanographic) conditions, are available in the Southern Ocean and even less in the MIZ (Derkani et al., 2021; see lines 115-117 of revised manuscript). We previously showed in Womack et al. (2022) that even in regions of 80-100 % sea ice concentration, the ice cover was not consolidated, as suggested by the high mobility of the analysed buoy (lines 123-124). It was further shown that the concentration based-definition (15-80 %) is inadequate to describe in the Antarctic MIZ, since the highly dynamic nature of the MIZ is maintained despite the high sea ice concentrations observed from remote sensing satellite products (see lines 125-127). This is due to the complex interactions between sea ice, polar cyclones and storm-generated waves (Vichi et al., 2019; Womack et al., 2022). Therefore, in this study we have, in its place, inferred sea ice type from the buoys' drift kinematics, spectral analyses and wind response (see Section 5.1, page 20). We have further clarified this conceptual principle in the revised Section 2 (see lines 191-197).*

**Page 7, The Eq. 4:** Further explanation is needed for the uncertainty of residual term estimation of mean ocean current.

*Answer: We thank the reviewer for this suggestion. The residual ocean current is the result of a linear model inversion, optimized through linear regression. Therefore, it inherits the uncertainties of the other known components, and accumulates part of the dispersion in the linear regression. To the best of our knowledge, there are no robust methods to determine under-ice geostrophic currents from remote sensing and extract proper statistics. In addition, like the estimates of the wind factor and turning angle, it may vary with the time period over which it is computed and the sampling interval of the transmission of the buoys. Time variations are not considered in our calculations (see lines 293-294). We have included a paragraph to discuss these uncertainties in Section 3, lines 298-309). As a further comparison (included as Appendix B, page 27-28), there is a remotely sensed product that can be used to compare with our estimate of the winter residual current, and check if it is compatible with the velocities observed in the ACC outside the sea ice at this longitude. We used data of current speed from Copernicus GlobCurrent database in the region of the buoy trajectories. This dataset combines the velocity field of geostrophic surface currents from satellite sensors recorded from 1993 to 2019 (Rio et al., 2014) and modelled Ekman currents, which include components from wind stress forcing obtained from atmospheric system and drifter data. In the figure below we have shown the mean currents for the period of the Winter buoys deployment (27 July 2019 – 5 October 2019). We extracted the distribution of the zonal and meridional currents between $-55^o$ S to $-56^o$ S (the mean of the buoys' latitudes) and between $0$-$26^o$ E (indicated by the cyan rectangular box). Again, we would like to reiterate that this product is a model, and it does have uncertainties, including the fact that geostrophic currents cannot be computed from altimetry under the sea ice, and therefore it is more likely that the velocities are biased towards the open ocean components far from the ice edge. Our estimated zonal and meridional under-ice currents fall within the range of the GlobCurrent distribution, although in the lower tail. Our aim was to demonstrate that at the latitude of the winter 2019 deployment it is more likely to find more intense ocean currents than at the latitudes of the 2017 winter deployment and 2019 spring deployments. This analysis indicates that currents can be expected to be even higher, which supports our hypothesis.*

[Figure]

*Figure: (left-hand side) The mean Copernicus GlobCurrent speed for the deployment period of the Winter buoys (27 July 2019 – 5 October 2019). The trajectories of the three Winter buoys are shown. The black contour denotes the AMSR2 0 % sea ice concentration on 30 September 2019 – the date of approximate austral sea ice maximum. The cyan rectangular box indicates the region used to compute the distribution of the zonal and meridional currents. (right-hand side) The distribution of the spatial mean of the GlobCurrent zonal and meridional currents. The vertical lines denote the magnitudes of the winter residual currents.*

**Page 12, Figure 4 :** Why does the square of AD rapidly increase in the first few days of calculation ?

*Answer: The square of the absolute dispersion rapidly increases as the figure is shown with the y-axis in log form, where the change from e.g. $10^{-3}$ to $10^{-2}$ are visually large. Additionally, the absolute dispersion is the square of the difference in position from the buoy's origin (deployment position) to each transmitted position. Therefore, it would increase as a power function as the buoys travel from 0 displacement (at time=0) to hundreds of kilometres squared.*

**Page 16, Table 2:** Confidence level of correlation coefficient?

*Answer: This was an oversight on our side. The confidence level of the correlation coefficient for the Pearson correlation was 95 % (line 312 of revised manuscript). We have now also reported the p-values for the Pearson correlations (lines 596-597), which were all statistically significant with a value < 0.05.*

**Page 17, Line 499:** "also statically non-significant in the wavelet spectrum"--This also depends on the length of the data time series.

*Answer: We agree with this comment by the reviewer, but the multiple time windows covered by the drifting data allowed us to gather a good confidence. For the period that the Winter buoys drifted (≈ 70 days), all three of the buoys indicated no statically significant power at the inertial frequency. This even includes ISVP 2 (Figure S10 of the supplementary material), which drifted for almost half the time (28 days) of ISVP 1 and ISVP 3. Therefore, we concluded that the advection velocity remained larger than the Coriolis forcing, throughout the Winter buoys' drift. We have amended this sentence in lines 666-667 of the revised version.*

**Page 21, Line 610** "Overall, the spring ice cover continued to move primarily as an aggregate over the » 30 day analysed period. This in agreement with Colony and Thorndike (1980) who showed a high coherency between

ice floes separated hundreds of kilometres apart at both the low and high frequencies"-- This literature citation may not seem very appropriate here. Whether the buoy array maintains consistent motion mainly depends on the deployment region, atmospheric and oceanic forcing, and the properties of sea ice itself.

*Answer: We acknowledge the genericity of our statement in this context, which does not add much information to the introduction. We thank for the suggestion, but have since removed this entirely from the discussion – Section 5.2.*

**Page 24, Line 689** "Thus, the magnitude of deformation, for similar spatial scales, may vary between different seasons"-- this literature gives a relatively complete description of the seasonal variation in sea ice consolidation and ice field deformation rate (although it is for the Arctic sea ice), and the initial ice conditions will affect the Rheology of the ice field in subsequent seasons:

> Lei R, Gui D, Hutchings J K, Heil P, Li N. 2020. Annual cycles of sea ice motion and deformation derived from buoy measurements in the western Arctic Ocean over two ice seasons. Journal of Geophysical Research: Oceans, 125, e2019JC015310. https://doi.org/10.1029/2019JC015310.

*Answer: We thank the reviewer for this suggestion. We are indeed conscious that not all the information acquired from the Arctic can directly be applied to Antarctic sea ice. We have however included a relation to this paper in lines 885-886 of the revised manuscript highlighting seasonal preconditioning.*

**Page 25, Line 719** "In summary, the present study highlights the need for a better understanding of the impacts of ocean currents and waves on the Antarctic sea ice cover to fully understand and quantify the effects of atmospheric forcing"-- it is very important to understand the impacts of ocean currents and waves on the Antarctic sea ice cover. However it is not just for understanding the effects of atmospheric forcing.

*Answer: We thank the reviewer for this comment. This statement was meant to be specific to the Antarctic marginal ice zone, where the atmospheric component is one of the main driver affecting the drift and the deformation (Vihma et al., 1996; Alberello et al., 2020). We have rephrased it in the revised manuscript, lines 916-918.*


General Comments:

**Comment 1:** The authors have presented a comparison of GPS trajectories of three winter and four spring buoys deployed in the marginal ice zone of Antarctica. The analysis primarily focused on three of the 7 trajectories (ISVPs 1, 3, & 4). Sea ice dynamics in the southern ocean is poorly understood and the authors have provided new data for the study of ice behavior in a very complex region. The effort necessary to collect this vital field data should not be understated. This is a very valuable avenue of research and contributions to Antarctic sea ice research is timely and important.

On the whole, I am struggling to see what new understanding we have gleaned from this data or this analysis. The authors report some degree of atmosphere-ice and ice-ocean coupling, but it is inconsistent and only loosely validated. The authors utilize a wide array of analytical methods: meander coefficients, absolute dispersion, relative dispersion, deformation rates, wind speed correlations, linear regressions, power spectral densities, and wavelet power spectra. The results from each analytical method is only briefly described, and it is not entirely clear to me how they complement each other.

As I understand the manuscript, the main findings could be summarized as follows:

- The primary tool for prescribing ice deformation behavior appears to be the identification of cyclones as nearby low pressure regions, which sometimes drive ice motion, and sometimes do not. When ice is more broken, there can be a better correlation to wind, such as in winter conditions. Otherwise, a solid ice pack does not respond the same way to wind.

- Cyclones can be responsible for the break up of ice.

*Answer 1: We thank the reviewer for this comment, which we interpreted in combination with Comment 3 below. The reviewer highlights the use of multiple diagnostics in our work, but a lack of coherence that clearly limited the message we wanted to convey. The reviewer gave very clear indications in their specific comments on how to improve, and in those answers, we proposed modifications that we hope will be satisfactory.*

*The main introductory aspects that we did not remark are the special features of the sea ice region we are measuring and describing in terms of its drift and deformation. And more specifically, how different they are from the Arctic (excluding embayments) and from the western Weddell Sea where most of the buoys are routinely deployed. There is an expectation that when sea ice attains 100 % coverage from space it becomes consolidated and subject to strong internal stresses. Earlier works done in East Antarctica and the Ross Sea demonstrated the penetration of waves into 100 % sea ice and (Kohout et al., 2014; Kohout et al., 2020; Alberello et al, 2022; Montiel et al., 2022). This is why we compared our data with the few buoy arrays deployed in similar conditions, namely the 2017 expeditions in the Indian Ocean sector (July 2017) and in the Ross Sea (April-May 2017).*

*With these limited datasets, we may not offer definitive conclusions, but we believe our main findings go beyond the two points indicated by the reviewer. This may likely have been caused from our distinct separation of the results from the discussion section (attended to in comment:* **Page 12, L347-348**). *These are the premises to our conclusions.*

*We show that strong winds, attributed to passing cyclones, maintain free-drift conditions, but this close relationship between sea ice and wind forcing is also modulated by the underlying currents (dampening inertial oscillations) and by the local ice conditions. In Womack et al. (2022), we presented how strong this relationship is > 200 km from the ice edge for a full deployment from winter to summer (see lines 122-125 of the revised manuscript). Here, we show that the window of consolidated conditions, in which Antarctic sea ice is more likely to present stronger internal stresses, is limited to the September-October period that we were unable to sample. In our spring measurements right after deployment, internal ice stresses were initially more significant, while the ice drift was still strongly related to wind forcing (page 20, lines 710-713 of revised manuscript). However, as increased air temperatures and wave activity caused the consolidated ice to melt and break-up into smaller ice floes (page 21, lines 700-703 of the revised manuscript), the drift became dominated by the initiation of the inertial response of sea ice (page 22, lines 724-727 of the revised manuscript). This difference between the Winter and Spring buoys is explained by their proximity to the ice edge and hence, the ACC, which enhanced wind-driven zonal drift for the Winter buoys (see lines 745 and 754-756 of revised manuscript). Deformation in winter continued to be strongly related to large-scale atmospheric forcing, while in spring there was a strong de-coupling between deformation and atmospheric forcing despite the expectation that a fractured sea ice would be more prone to follow the wind forcing (see lines 828-829 of the revised manuscript). This difference is highly significant as it highlights Antarctica's complex and varying ice cover. With the incorporation of additional buoy arrays across different regions around Antarctica, our deformation analysis is one of the most extensive in the Southern Hemisphere.*

*This manuscript uses a wide variety of metrics to analyse ice drift and deformation in the poorly sampled and poorly understood Antarctic MIZ. We provide unique data and perform many analyses to generate insights, which all lead to the same conclusions for each season (pages 25-26 of revised manuscript). This manuscript leads to important conclusions as well as recommendations for in situ oceanic observations around Antarctica and thus may be of significance to inform future buoy deployments (page 26, lines 922-926 of revised manuscript).*

*We would like to note that according to our observations from multiple winter cruises in the MIZ, cyclones are not directly responsible for the break-up of sea ice (as opposed to Arctic conditions, e.g. Granskog et al., 2016). It is rather their storm-generated waves near the ice edge that make the ice more mobile as they propagate into the ice cover, possibly re-establishing the separate mosaic of individual floes previously cemented together (see lines 47-51 and 700-702 of revised manuscript). This allows the ice cover to be more susceptible for drift and deformation by winds and ocean currents (added to lines 702-703 of revised manuscript). Additionally, the thermal effect of the warmer maritime air injected by cyclones cannot be excluded (e.g. Vichi et al., 2019), and, as stated above, the seasonal change in air temperatures appear to have a significant role in ice melt and break up during the spring season, and hence changing the response of the ice cover to wind forcing.*

**Comment 2:** There are a number of mathematical questions I have also detailed in my specific comments that I think should be addressed as fluctuation magnitudes are frequently mentioned, but those magnitudes are only reported qualitatively, and result from normalizing the time variation of certain distances by different initial separations. I find it difficult to see how these data, especially quantifications of dispersion or deformation will be readily comparable as there seems to be caveats attached to each buoy trajectory.

***Answer 2:*** *We apologise if we are not interpreting the reviewer's comment correctly. We answered all the specific questions, and we hope this addressed the concerns that originated this comment. We think the reviewer refers to the fluctuation of magnitudes of the dispersion/deformation diagnostics we computed. They are indeed simple normalization based on initial separations, but nonetheless they are objective measures and not qualitative. Additionally, all of these measures have been backed up by quantitative methods such as power spectral analyses*

*(Figures 6 and 8) and correlation coefficients (Table 2). Our choice for the use of dispersion statistics was based on the quasi-linear deployment of both buoy arrays, that did not allow for the computation of total deformation using the combined strain rates. We further explain in Answer 7 below for this choice, where we also present a further analysis using the TSEs recently proposed by Aksamit et al. (2023).*

*In reference to the caveats attached to each buoy trajectory, buoys follow ice floes on which they are deployed on, therefore we have limited control over where the data will be available (Aksamit et al. 2023). Additionally, the paucity of observations in the Antarctic (and even more in the MIZ) has meant that many of the drivers of the Antarctic sea ice seasonal evolution are poorly understood – specifically sea ice drift, deformation and ice type in the MIZ (see lines 115-117 of the revised manuscript). Majority of our scientific knowledge of sea ice drift and deformation has been derived from pack-ice conditions (e.g. Vihma et al., 1996; Uotila et al., 2000; Lindsay, 2002; Doble and Wadhams, 2006; Ackley et al., 2015). Therefore, our knowledge of metocean properties in the Antarctic MIZ is still incomplete, and hence our abilities to accurately model them remains limited. It is important that we analyse these buoy arrays as they do supply key information at both the synoptic and sub-daily time scales. Additionally, our results of sea ice drift and deformation have been readily comparable to prior literature. All ice drift (Section 5.1) and deformation (Section 5.2) results have been comprehensively discussed and related to other research in the Antarctic, including the deformation proxy which highlighted the de-coupling phenomenon that has been noted by prior Antarctic studies (lines 800-805 of revised manuscript). Lastly, as reported in lines 207-210 of the revised manuscript, we did note the uncertainty of whether the Winter buoys remained within the ice edge and that caution was taken when analysing the dataset. ISVP 1 was additionally discussed separately in lines 775-786 of the revised discussion.*

**Comment 3**: It would be insightful for the authors to show specifically what new information we have achieved through this research. It would be especially beneficial if the authors could justify the necessity of a field campaign like this by showing how this information is not readily available from other data sources, such as gridded sea ice products, remote sensing products, or geostrophic ocean currents. So far, the authors make no comparison to other sea ice products, besides a demarcation of sea ice concentration boundaries.

*Answer 3: This comment is legitimate, but partly in disagreement with the introductory Comment 1 made by the reviewer on the limitedness and complexity of collecting data from the Antarctic marginal ice zone. Not to mention the remarkable ability that we had to conduct similar data collections (albeit limited by the logistics and weather-related issues of each expedition) in two different seasons of the same year. We thus agree with the comment made below about the title that the year 2019 should be explicitly included.*

*This expedition generated an enormous set of comprehensive data from the physics to the biogeochemistry of the MIZ in the Atlantic sector that is available to any researcher on the SCALE community repository on Zenodo (https://zenodo.org/communities/scale_south_africa/). We do not think we need to justify the necessity of this, because it is self-explanatory. We reckon that we must explain clearly what new information has been generated by the analysis of these buoy data, not the necessity of the buoy data themselves. The reviewer mentioned the lack of comparison with other sea ice products, maybe unconsciously lacking to realise that this would be reverting the natural process of Earth Observation (modelling) products' validation. The data we collected and presented, as well as the information we derived from multiple diagnostics, should be used downstream by the community to assess the quality of the products derived from various algorithms and models, not the other way around. Despite the common misnomer, we would like to remark that remotely sensed "data" are ultimately modelling products with their own uncertainties and limitations.*

*We do thank the reviewer since this comment made us aware that not all readers may be knowledgeable of the very limited set of in situ data used for the validation of these remotely-sensed datasets in the Antarctic, including both altimeter, radar, and passive microwave products. These products have been extensively assessed in the Arctic thanks to the continuity of direct in situ observations from buoys, expeditions and long-term observatories naturally bordering the Arctic ocean (see figure below from OSI-SAF validation report; Lavergne and Down, 2022). Yet, in the Antarctic, the number of validating buoys routinely used is very limited to the ones available in*

*the IPAB database (https://www.ipab.aq/), which the reviewer would quickly realise that they mostly cover the Weddell Sea on a routine basis (Lavergne and Down, 2022), and not in the region of our study (red box in figure below). Our data very likely represent 70 % of the buoys in the eastern Weddle Sea and East Antarctica sectors that can be confidently assumed to be drifting on sea ice. We have added this further information in the revised manuscript, lines 117-120.*

[Figure]

*Figure: Buoy coverage 2011-2020 for the Northern (a) and Southern (b) Hemisphere. The colour scale indicates the time inside the decade. Data records from all seasons are plotted. Figure 2.3 and caption from OSI-SAF validation report (Lavergne and Down, 2022).*

*We included, in the analysis of our in situ data, the satellite information that we consider reliable and usable at this stage in the region. The ability of sea ice drift products to produce drift fields, in comparison to drifting buoys, will be attended to in comment 5. We have included better clarification in the revised manuscript as to why this caution should be exercised in the Antarctic MIZ (see lines 191-197). Please note that this is different from the use of atmospheric reanalyses, which have been independently validated in the expeditions we conducted in the past in the region (e.g. Vichi et al., 2019; King et al., 2022).*

*The following details our full reasoning, and we will include an extract in the revised discussion:*

*The use of satellite products is known to be limited in their application to the broad Antarctic MIZ, as they are less reliable in the Southern Ocean, where ice type is less related to the concentration value (Vichi, 2022). This is because very little field data of metocean (meteorological and oceanographic) conditions are available in the Southern Ocean and even less in the MIZ (Derkani et al., 2020; lines 115-117 of revised manuscript). The amount of in situ data available is not sufficient enough to be used for regular validation (Aaboe et al., 2021), which has had drawback effects for prediction models which are impaired by significant biases in the Southern Ocean (e.g. Yuan, 2004; Li et al., 2013; Zieger et al., 2015). This has been included in the revised manuscript (lines 117-118 and 191-197). We previously showed in Womack et al. (2022) that even in regions of 80-100 % sea ice concentration (AMSR2 and SSMIS), the ice cover was not consolidated, as suggested by the high mobility and strong inertial signature exhibited by the analysed buoy (see lines 120-125 of revised manuscript). This is in contrast with buoys deployed in the Weddell Sea in Autumn (e.g. Doble et al., 2003). It was further shown that the concentration based-definition (15-80 %) is inadequate to describe the Antarctic MIZ, since the highly dynamic nature of the MIZ is maintained despite the high sea ice concentrations observed from remote sensing*

*satellite products (lines 125-127 of revised manuscript). This is due to the complex interactions between sea ice, polar cyclones and storm-generated waves (Vichi et al., 2019; Womack et al., 2022). If we cannot rely on satellite products in the Antarctic, we cannot rely on gridded sea ice products to determine high-resolution ice characteristics. For this reason, we do not use satellite ice concentrations other than to estimate the ice edge, and hence, we have (in place of remotely sensed sea ice concentration) inferred sea ice type from the buoys' drift kinematics, spectral analyses and wind response (see Section 5.1, page 20).*

**Comment 4**. There is also no validation of their ocean current estimates.

***Answer 4:*** *We have separated this comment from the previous one, since it deserved a more accurate treatment and we failed to do so. The current is estimated as the residual parameter of the least square regression used to estimate the wind factor and turning angle. As such it carries multiple uncertainties that are not directly quantifiable. We are aware of the absence of in situ observations for under-ice ocean currents in the Antarctic MIZ (see for instance Alberello et al., 2020, where this issue was initially remarked for the Antarctic). There are contingent logistical reasons for this, since current meter moorings can only be established in consolidated pack ice, and ship-based ADCP instruments are unusable within the ice field. To the authors' knowledge, the only in situ observations of underlying ocean currents for the Antarctic were reported by Geiger et al. (1998). These observations were measured in the western Weddell Sea, along the coast, where the Antarctic Coastal Current is located and land-fast ice and pack ice conditions dominate. These ice types do not have an equivalent spatial morphology to MIZ ice conditions (Vichi, 2022), and the coastal current incurs significantly different driving mechanisms. Therefore, we cannot validate our residual currents to in situ data. This knowledge now has been included the revised manuscript (lines 299-309).*

*The importance of the residual current here is to explain the difference between winter 2019 and winter 2017 deployments, in which we observed evident inertial oscillations (Alberello et al., 2020; Womack et al., 2022) that are absent along the whole trajectory of the 2019 buoys, despite the similar MIZ conditions (see lines 753-756 of the revised manuscript). Following the reviewer's comments, in this case there is a remotely sensed product that can be used to compare with our estimate of the winter residual current, and check if it is compatible with the velocities observed in the ACC outside the sea ice at this longitude. We used data of current speed from Copernicus GlobCurrent database in the region of the buoy trajectories. This dataset combines the velocity field of geostrophic surface currents from satellite sensors recorded from 1993 to 2019 (Rio et al., 2014) and modelled Ekman currents, which include components from wind stress forcing, obtained from atmospheric system and drifter data. In the figure below we have shown the mean currents for the period of the Winter buoys deployment (27 July 2019 – 5 October 2019). We extracted the distribution of the zonal and meridional currents between -55$^o$ S to -56$^o$ S (the mean of the buoys' latitudes) and between 0-26$^o$ E (as indicated by the cyan rectangular box). Again, we would like to reiterate that this product is a model, and it does have uncertainties (Derkani et al., 2021), including the fact that geostrophic currents cannot be computed from altimetry under the sea ice. Therefore, it is more likely that the velocities are biased towards the open ocean components far from the ice edge. Our estimated zonal and meridional under-ice currents fall within the range of the GlobCurrent distribution, although in the lower tail. Our aim was to however demonstrate that at the latitude of the winter 2019 deployment it is more likely to find more intense ocean currents than at the latitudes of the 2017 winter deployment and 2019 spring deployments. This analysis indicates that currents can be expected to be even higher, which supports our hypothesis. This has been included in the revised manuscript as Appendix B and Figure B1 (page 27-28).*

[Figure]

*Figure: (left-hand side) The mean Copernicus GlobCurrent speed for the deployment period of the Winter buoys (27 July 2019 – 5 October 2019). The trajectories of the three buoys are shown. The black contour denotes the AMSR2 0 % sea ice concentration on 30 September 2019 – the date of approximate austral sea ice maximum. The cyan rectangular box indicates the region used to compute the distribution of the zonal and meridional currents. (right-hand side) The distribution of the spatial mean of the GlobCurrent zonal and meridional currents. The vertical lines denote the magnitudes of the winter residual currents.*

**Comment 5:** What sort of information does this expedition gives us that we couldn't get from a sea ice drift product (e.g. NSIDC or OSISAF)? Do Lagrangian trajectories generated from one of these drift products recreate similar dynamics, even at a lower resolution?

***Answer 5:*** *Thank you for this question, we believe we gave the full explanation in our Answer 2 above. We would argue that as our data have been collected and their diagnostics analysed; only now it is possible to start using the existing products for deriving similar diagnostics. This was not the aim of our work, which rather focused on determining ice drift and deformation in 2019 Winter and Spring, which can be used to inform further analysis using models and Earth Observation products.*

*Indeed, the drift product from OSI-SAF is usable, in the limits of its resolution. We demonstrated this in Johnson et al. (2023) for the Spring buoys (their Fig. 2). The mean geodetic distance between the daily in situ trajectories and the corresponding simulated ones were within the range of the OSI-SAF grid resolution of 62.5 km. Thus, one could use satellite products to reproduce the daily drift and hence the deformation of sea ice at that scale, with a spatial uncertainty of more than 50 km, which is one of the several scales we analysed in the manuscript. However, remotely sensed products are more suited for large-scale and long-term drift and deformation patterns (Kwok et al., 2017). Satellite data have a low temporal resolution of one day, and thus would not be able to resolve any of the sub-daily variabilities related to synoptic events (Heil et al., 2001). Buoy data instead contain the sub-daily and true daily information on Lagrangian ice drift and deformation (Heil et al., 2009). These higher-frequency data are vital to differentiate atmospheric and oceanic forcing as well as to define the inertial signal in ice drift and deformation. In this manuscript, we show that the inertial response significantly influenced our analysis of ice drift (Figure 5; Figure 6, Figure 7), and deformation (Figure 8). Additionally, Heil et al. (2009) and Womack et al. (2022) reported that large-scale atmospheric forcing is the primary forcing mechanism of Antarctic sea ice drift, with secondary effects from the initiation of the inertial response of sea ice. Heil and Hibler (2002) have also shown that it is necessary to include the sub-daily drift and deformation in order to fully describe sea ice evolution using dynamic and thermodynamic sea ice models. Thus, buoy data provide vital information that satellite products cannot. A short reasoning has been included in the revised manuscript, lines 145-147.*

**Comment 6:** I think that the title is a bit too broad. This is one year of data and it should be clear that this is a case study and does not claim to report on either average or atypical behavior. For example "**A contrast in sea ice drift and deformation between winter and spring 2019 in the Antarctic marginal ice zone"** would be more appropriate.

*Answer 6: We acknowledge this suggestion by the Reviewer, and it has been changed in the revised manuscript (page 1). The reason we chose a more general title is that we compare our results with other buoys from different periods and regions, but we acknowledge that it is still a limited dataset. More experiments like this one should be done before we can generalize to the whole Antarctic region and use satellite products more reliably.*

**Comment 7:** There is limited discussion on why other diagnostics were not also compared (which should be in the methods and not in the results). Specifically, why did you not use the common Green's theorem based buoy-array diagnostics, or the relatively untested Trajectory Stretching Exponents

*Answer 7: Thank you for this comment. While sea ice deformation computed from divergence, shear and total deformation is the most common approach to determine differential drift of sea ice; it is obtained from analysing buoy arrays whose vertices form polygons or triangles and computed using discretised contour integrals (Green's theorem) at each time step (Hutchings and Hibler, 2008; Itkin et al., 2017; Aksamit et al., 2023). This has been remarked in lines 76-81 of the revised manuscript. To compute the strain rates using this approach, pre-determined polygon areas need to be tracked (Hutchings et al., 2012), or all triangles formed by the buoys must not have small angles i.e. $< 15^o$ (Itkin et al., 2017). However, as previously reported in lines 332-335 on page 9, both the Winter and Spring buoys were deployed in a quasi-linear buoys array geometry, thus all triangles formed by the buoys had small angles. This would have given unreliable calculations of the strain rates as explained by Itkin et al. (2017), and the reduction in the accuracy from a large array to only two buoys is unknown (see Alberello et al., 2020). We have used these arguments in the revised version (Section 3.4, page 9) to better clarify why we could not analyse the total strain rate tensor to compute sea ice deformation.*

*We agree with the reviewer that Trajectory Stretching Exponents (TSE) is an insightful method to describe local deformation of sea ice, as it is not limited by the need for multiple buoys or the preservation of triangle shapes which is difficult in a mobile MIZ. We were aware of the TSE method in the ocean, but the application to sea ice was still under discussion and unpublished when we submitted. The TSE indeed provides a more accurate analysis than the one we obtained with single-particle dispersion. This strengthens our analysis for each season since we can better see the compression and stretching associated to the cyclones (see Figure below). The TSEs increase in association with the passage of cyclones, quite often before the cyclone core is closest to the buoy's location. This is not surprising given the large spatial scales of these atmospheric features (e.g. Hepworth et al. 2022 and references therein). In the revised manuscript, we have included TSEs (see Fig. 4e-f, page 13).*

*We would like to note that we have decided not to include $\overline{TSE}$, since our proxy for deformation (Rampal et al., 2008, 2009) is readily comparable with other Antarctic literature (e.g. Geiger et al., 1998; Heil et al., 2008, 2009, 2011) and Arctic literature (Lindsay et al., 2003; Marsan et al., 2004; Rampal et al., 2008, 2009; Hutchings et al., 2011), see Section 5.2 of the manuscript.*

[Figure]

*Figure: (a) Time series of the TSE for the 2019 Winter buoys. (b) Same as (a) but for the Spring buoys. The stars are the same as in Fig. A1.*

Specific Comments:

**Page 2, L75-76:** Haller (2015) actually does not say anything about the ability of absolute dispersion to provide a signature of circulation or structure organization. That reference only states that absolute dispersion is a frame-indifferent Lagrangian diagnostic. Haller and Yuan (2000) actually advocate for relative dispersion over absolute dispersion as it has a more direct connection to stable and unstable material lines which correspond with repelling and attracting Lagrangian coherent structures, such as around zones of large shear or material separation/aggregation. Encinas-Bartos et al., 2022 also detail that absolute dispersion does not capture elliptic LCS (e.g. eddies) as it "lack(s) direct physical connection to material deformation in the fluid." These citations seem to be in direct conflict with your claim that absolute dispersion can be used to represent flow structures. As there is no direct connection to material deformation or deformation structures, can you comment on why this is a reasonable way to quantify deformation of sea ice?

***Answer:*** *We acknowledge this comment by the reviewer. The citation to Haller (2015) was misplaced and has been removed. Although Haller and Yuan (2000) may advocate for the use of relative dispersion over absolute dispersion, we (along with LaCasce, 2008 and Lukovich et al., 2017) find it important to analyse both single- and multi-particle dispersion statistics in order to provide a full description of ice floe evolution (lines 88-89 of revised manuscript). Therefore, following the approach by Lukovich et al. (2017), we applied both methods to quantify ice drift and deformation in the Antarctic MIZ (Section 3.4, page 9). Additionally, while Encinas-Bartos et al., (2022) reported that they find that absolute dispersion does not capture elliptic Lagrangian coherent structures, Lukovich et al. (2017, 2021) and Womack et al. (2022) have found it to be useful to describe drift dynamical regimes during the deployment of buoys drifting with sea ice. Womack et al. (2022; their Figure 9) related the sub-diffusive (elliptic) regime to periods when cyclones passed over the Antarctic MIZ, resulting in compression of the ice cover by the meridional winds. More notably, they also related the dates of these "trapping" events to the dates of increased power at the inertial frequency, which coincided with the dates of the three large loops in the analysed drift trajectory. However, in this manuscript we do not characterise the dynamical regimes but rather described the response of the ice cover in each season to the passage of cyclones (and inertial oscillations), using absolute dispersion as one of the approaches. Absolute dispersion was thus used to characterise ice motion in the zonal and meridional directions, but we recognise that the TSE analysis recently proposed does enhance the capability to distinguish compression and stretching, which is not possible with the absolute dispersion. Fluctuations in the ice velocity field can be attributed to deformation processes or internal ice stresses.*

**Page 3, L87-88:** Can you clarify this sentence? What do you mean total deformation as a cluster? How would we potentially want to determine total deformation, and why wouldn't that work?

***Answer:*** *Thank you. As reported above, sea ice deformation is commonly reported using the strain rates (divergence, shear and total deformation), of a triangle or polygon areas. However, not all buoys are deployed sufficiently well to determine the total strain rate tensor, as reported above. Total deformation as a cluster means the total deformation computed by the combined strain rates over the analysed area or grouping area of buoys.*

*This is fully addressed in the methods section, but have removed the ambiguity of this sentence in the introduction when rearranging it to include additional information on deformation and TSE (see pages 2-3).*

**Page 4, L136-137:** Can you be more precise about the scale of meteorological systems you are trying to relate to? Do you have faith in the reconstruction of mesoscale features? Synoptic-scale features? Can you provide a reference to the reliability of either in this region? The dependency on temporal and spatial resolution of ERA5 in the southern ocean is still actively being investigated (e.g. Zhong et al., 2023)

*Answer: Thank you for this suggestion. In this manuscript, we analyse synoptic cyclones, which have a radius ranging between 500 to 2000 km (Hoskins and Hodges, 2005; Uotila et al., 2011). As to the reliability of atmospheric variables in the Southern Ocean, King et al. (2022) used three drifting buoys in the Weddell Sea to evaluate the performance of ERA5 and ERA-Interim. They showed that these two products captured the temporal variability of surface pressure and near-surface temperature well, with small biases in the 10-m wind speed. Additionally, Vichi et al. (2019) reported that the atmospheric variables measured along the SA Agulhas II's cruise track in 2017 ($\approx 30^o$ E) compared very well with the corresponding reanalysis data from ERA5, especially for mean sea level pressure. The near-surface temperature and wind velocity were affected by a linear bias that can be corrected. We could not repeat this validation this time since the ship meteorological data were sent to the WMO for assimilation into the global reanalyses, and the assessment would not be independent. In addition, as reported in lines 154-155, there were reliability issues with the ISVPs' meteorological data, and the Trident buoy only recorded air temperature and its GPS position (line 168). The manuscript has been revised in lines 183-184 to include these two additional points.*

**Page 5, L148-149:** What edge is heterogenous and fragmented? The 0% or 15% line?

*Answer: The 0 % line. This has been clarified in the revised manuscript, line 199.*

**Page 5, L151-152:** Specifically what findings were only marginally affected by this choice? Please rephrase.

*Answer: Thank you for this suggestion. It has been attended to in the revised manuscript, line 203. Womack et al. (2022) reported that their results of the buoy's distance from the ice edge in the Antarctic were only marginally affected by this choice, with a maximum difference between the 0 % and 15 % SIC of less than 50 km.*

**Page 5, L158-163:** Can you report how many days you got useable in-ice data for all your GPS sampling? This seems potentially inconsistent across the manuscript.

*Answer: Thank you for this suggestion. We have included the number of days in the revised Table 1, page 5.*

**Page 6, L176-177:** This phrasing is a bit awkward. You had a small meteorological dataset to investigate because you had very short trajectory data, so you were able to do this manually, correct?

*Answer: The Reviewer is correct. We will rephrase the sentence (see lines 229-230 of revised manuscript), although the length of the trajectories is not short according to the typical deployment duration of ice-tethered Antarctic buoys in the MIZ.*

**Page 6, L189:** Position error <5 m?

*Answer: The GPS position of all seven buoys was communicated via the Iridium system, with an accuracy of < 5 m.*

**Page 6, L191:** Why different sampling rates for the Trident and ISVPs?

*Answer: The ISVPs were deployed by the South African Weather Service. ISVP 1 transmitted data every 30 minutes but was re-sampled to 1 hour in order to be comparable to the other ISVPs (see Table 1). The Trident buoy was built by the Electrical Engineering Department at the University of Cape Town, and was designed to only transmit data every 4 hours. We decided to keep the higher temporal resolution of the ISVPs since it provided us with more accurate data on ice drift (see lines 244-245 of revised manuscript).*

**Page 6, L191-192:** If you can test the lower resolution, why was this only tested for one buoy?

*Answer: We acknowledge this comment by the reviewer and therefore has been tested for all spring ISVPs and not just ISVP 4. The same conclusion can be made. For example, see Figure S5 in the supplementary material, where the Spring buoys' meander coefficient and TSEs were computed using the Trident's 4-hourly time interval.*

**Page 6, L199-206:** Can you please provide plots of the raw timeseries of u and v for your buoys? It sounds like you have minimal noise and did not have to undergo any smoothing or cleaning of the data, which is not typically the case, and should be celebrated!

*Answer: Only ISVP 1 was resampled from 30 minutes to 1 hour. The figure below shows ISVP 1's raw time series of u and v, and can be included in the supplementary material. During the resampling in python, the mean method was used and thus some of the extreme values were reduced. However, much of the time series remains the same during the resampling. For the rest of the buoys: ISVP 2 and ISVP 3 had a few duplicated time stamps, and the Trident buoy had a few missing data points that were filled so that the PSD and wavelet could be conducted. This is stated in lines 242-244 of the revised manuscript. However, none of the buoys underwent a smoothing process.*

[Figure]

*Figure: The (a) zonal and (b) meridional velocity components of ISVP 1, with a 30 minute sampling frequency (on the left axis), and 10 m wind from ERA5 reanalyses (on the right axis). The gold stars denote when each cyclone core was the closest to the Winter buoys.*

**Page 7, L213:** Missing parenthesis

*Answer: Thank you. This has been attended to.*

**Page 7, L235-240:** How close are c_u and c_v to the nearby geostrophic currents reported in the MIZ or just outside the MIZ (e.g. AVISO products)? In off-shore locations, there should be a reasonable connection to the buoy velocity, and the wind influence in the open water would be represented by Ekman velocity at a given latitude, correct? How close is theta to that used to calculate the Ekman velocity for open water?

*Answer: As discussed in Comment 4, there are no in situ observations of under-ice currents in the Antarctic MIZ. We have now included an analysis of Copernicus GlobCurrent data, which is a blended altimetry product that reconstructs the geostrophic and Ekman currents (see Appendix B). The product also gives currents under ice, which are however untested and as uncertain as our estimates. The open ocean currents have been compared with the 2019 Winter buoys, indicating that our estimates are on the lower range, which is compatible with under*

*ice momentum reduction. As the Spring buoys were further away from the ice edge, no current data could be used. Ekman velocity of surface currents is at an angle of $45^o$ to the left of wind direction in the Southern Hemisphere. This large difference between the angle of Ekman velocity and the turning angle is because sea ice forms a natural layer between the atmosphere and surface ocean. This leads to a profound influence on the exchanges momentum between the atmosphere and ocean. However, all of our reported turning angles fall within the commonly reported range of $0^o$-$30^o$ negative in the Southern Hemisphere (Leppäranta, 2011). This range has been included in the revised manuscript, lines 58-59 and 592-593.*

**Page 8, EQ 5:** Are you calculating this for the same t=0 for all buoys in each season, or from each buoys time of deployment?

*Answer: Thank you. Absolute dispersion, relative dispersion and the deformation proxy were computed from the time at which all buoys in each season began transmitting together until one of the buoys stopped transmitting (see lines 468-469, and Figures 3 and 4). This has been clarified in the revised methods section (lines 330, 359-360) as well, rather than just in the results section.*

**Page 8, L271-275:** I think this should be stated earlier in the methods.

*Answer: We acknowledge this suggestion by the reviewer, however these lines only affected our analysis of deformation (proxy) of the ice cover. Absolute and relative dispersion only require buoy pairs. As stated above, we could not compute the full strain rate because of the buoys' quasi-linear deployment orientation.*

**Page 12, Section 4:** How was the choice of 30 day windows for absolute dispersion and sigma_D made? Why is this a suitable value, and what happens at shorter and longer timescales? Presumably the cyclones are much shorter than this. It currently appears to be an arbitrary choice.

*Answer: Thank you for this comment. We did not choose this $\approx$ 30-day window in either season. As reported in lines 468-469 and Figures 3 and 4 (and now further clarified in the methods section) we were limited by the period for which all buoys, in each season, transmitted data together. The $\approx$ 30-day window was the longest period we could analyse for winter and spring, and since both seasons happened to have the same duration they were comparable. Like any other approach, e.g. TSEs or strain rates, dispersion statistics are effected by the time frame analysed, due to local weather and ice conditions. Therefore, shortening or lengthening the analysed period would affect the diagnostics. In our case, we chose to use the longest time available.*

**Page 9, L309-310:** This is a bit awkward phrasing because you actually identified your cyclone by this pressure minima.

*Answer: Thank you for this comment. But as detailed in our methods section 3.1 (lines 228-238 of revised manuscript) we identified the large-scale cyclonic features using a visual inspection of ERA5 mslp and 2-m air temperature fields to find cyclones with core pressures < 970 hPa and < 1000 km from the positions of the buoys. Once the eight cyclones in winter and seven in spring were found, we then plotted the time series of the collocated ERA5 variables. The stars represent the dates when the cyclone cores were the closest to the buoys. For this reason, the time series of the mslp may not drop to < 970 hPa during the passage of a cyclone, since it is not the core's mslp.*

**Page 9, L309-317:** Can you expand on the significance of these conditions?

*Answer: Before we answer this question, we would like to note that this figure and explanation has been moved to Appendix A. All atmospheric conditions were comprehensively related to ice drift and deformation in the discussion section. All drift and deformation phenomena have been related to the environmental conditions. See lines 698-703; 715-716; 729-732; 740-743; 791-793; 840-842 etc. Additionally, throughout the results section, the results were associated to cyclones and in spring, also the air temperature.*

**Page 10, L326-330:** Again it would be more helpful to report how many days you were able to use the trajectory data instead of the dates.

*Answer: We thank the reviewer for this comment. As stated above, we have included a column in Table 1 with how many days the buoys were in ice.*

**Page 10, L330-331:** I don't think a reference to Figure 2a is appropriate here as we don't actually see the large turns that are being described, just the coincident cyclones.

*Answer: We thank the reviewer for this comment. It has been removed.*

**Page 11, L326-336:** It would be very helpful if you could describe the features of Figure 3, such as the coloring, black lines, gray lines. It looks like there is no ISVP2 termination point (is that was the X's are?). Why do the tracks extend beyond the ISVP1 and ISVP3 X's?

*Answer: Thank you for this suggestion. We have included an additional point in revised Figure 2 (page 12) to indicate ISVP 2's termination point. However, in the figure caption, we comprehensively detailed all the features that were shown. There is also a time gradient colouring for the buoys trajectories, with a colour bar shown (Fig. 2a and b). The gantt chart (Fig. 2c) additionally indicates the deployment duration of all buoys, with also their starting and ending longitudes. The trajectories extend beyond the X's because the X's denote the position of ISVP 1 and 3 on the date of approximate austral sea ice maximum (30 September 2019) and the buoys continued drifting until 5 October 2019 (end of analysis period; see Table 1).*

**Page 11, Figure 3:** I think this figure could use some improvement, particularly some labels.

*Answer: Thank you for this suggestion. We have decided not to add any labels of the buoys' start and termination dates to the figures since they are already detailed in Table 1, and are additionally shown in now revised Fig. 2c (operation dates). All the buoys had different start and end dates and therefore the figure would have looked too cluttered. There is a legend for the colours of each buoy, with a full description in the figure caption below. However, as described above, ISVP 2's termination point is now included.*

**Page 12, L347-348:** Is this because of a change in the ice, or because the buoys moved into a region with a different dominant flow feature?

*Answer: Thank you for this comment. We explain and justify why the meander coefficient decreased in the discussion at lines 748-752. Due to the confusion of the reviewer, we have included a new comment to the start of the results section (379-382 of the revised manuscript) explaining that we first present all results, and then only in the discussion do we discuss, analyse and correlate all results comprehensively.*

**Page 12, L359-366:** This level of information from absolute dispersion does not contribute to the manuscript.

*Answer: Thank you for this comment. We described the absolute dispersion here, but discussed in in Section 5.1, where we correlate it to the trajectory shapes of the buoys in each season. Additionally, we have now merged the analysis of the absolute and relative dispersions and separated the deformation proxy.*

**Page 13, L373-374:** Quantify "more variability". Do larger fluctuations just scale with the order of magnitude difference in the values? It's hard to see meaningful differences in fluctuations when you are using a log plot positioned at different y-values.

*Answer: We thank the reviewer for the comment. In this section, we first describe the figures. We discuss and correlate these results with other quantitative measurements in the discussion (Section 5.2). We have added a*

*comment to the start of the results section explaining that the results are only presented here and later discussed in the discussion section (5.1 and 5.2).*

**Page 13, L382-383:** I don't think you can claim that the buoys moved more coherently as an aggregate. In both winter and spring the ice could theoretically have the exact same values of divergence corresponding to a linear velocity field, but the larger initial separation would give you lower relative dispersion by nature of the larger denominator.

*Answer: We acknowledge this comment by the reviewer. For this larger spatial scale, the Spring buoys do move more coherently than the Winter buoys of a smaller spatial scale. Again, we present the results here. In Section 5.2, we discuss the spatial dependence of deformation, using the Winter and Spring buoys as well as two additional buoy arrays that were available to us from different regions around Antarctica (see Figure 9, page 24).*

**Page 13, Section 4.3:** When comparing the range of values and fluctuations, I think you need to provide quantitative information. It is not easy to see the relative differences between fluctuation ranges on these log plots.

*Answer: Thank you for this comment. This has been attended to in the above comments, but we did previously compute a PSD of the deformation during winter and spring to provide additional quantitative information (see Figure 8, page 23).*

**Page 13, L383-384:** Is this larger meridional fluctuation not just related to the smaller meridional separation that you started with?

*Answer: Yes, this is true but also because the Spring buoys drifted northwards under the action of the inertial response of the ice cover, which lead to a significant decrease in the meridional separation (see line 720-721 of the revised manuscript).*

**Page 13, L386:** Figure 3b?

*Answer: Thank you for this comment. This has been clarified in the revised manuscript.*

**Page 13, L398-399:** Same statement as above about quantifying the difference in fluctuation ranges.

*Answer: Thank you. We have attended to this comment above.*

**Page 14, Figure 5:** No legend for cyclone stars

*Answer: We acknowledge this comment from the reviewer. In the caption for now revised Figure A1 (from old Fig. 2) we detail what each coloured star represents. Additionally, in Figure 5's caption we note that the star symbols are the same as in Figure A1. This is approach is done for all figures as to not be repetitive.*

**Page 13, L403-404:** I am not convinced this indicates the winter ice cover was deformable in response to winds. There also appear to be just as many cyclones occurring where there is not a significant fluctuation in sigma_D. A deeper analysis is required to explain why it is reasonable for only these events to result in such a sea ice response, and why we can causally link that to the wind. What else is happening to the sea ice at this time?

*Answer: We acknowledge this comment by the reviewer. We understand that this is mostly a stylistic issue in terms of the manuscript organization. As reported in the comments above, we present and explain the figures in the results section (Section 4), but only discuss, analyse and correlate all results in the discussion (see Section 5). We comprehensively discuss why it is reasonable to correlate the deformation with wind forcing in lines 838-850 of revised manuscript, which additionally includes a PSD of the deformation (Figure 8). This strongly links the deformation to large-scale atmospheric forcing. We additionally suggest that this was due the steady source of*

*low-frequency energy from the ACC, which was in strong coherence with the wind. As suggested earlier, we have now ensured that the reader will know that the results from Section 4 are comprehensively discussed in Section 5.*

**Page 14, L405-406:** Can this change in fluctuation magnitude be attributed to the difference in L0 that you are dividing by? That is, the oscillations of all buoys all similar, but the larger distance between them "dampens" the effect?

*Answer: Thank you for this comment. We discuss this in full in the discussion section (5.2) in lines 795-798. Here, we showed with the utilisation of the PSD of the deformation (Figure 8), that these changes were due to a combination of the passage of cyclones and inertial oscillations.*

**Page 15, Figure 6:** Can you confirm that this is a raw, unfiltered velocity? And computed from what sampling rate? As well, it would seem to make more sense to report the rotation angle theta and the rescaled/rotated velocity contribution from the 10 meter wind, not the ERA5 output, as that is what you are suggesting is "driving" the buoy motion.

*Answer: As stated above, ISVP 1 has been resampled to 1 hour, while the other buoys are plotted with their original sampling rate (including the buoys in the supplementary material).*

*We hope that the following answers the reviewer's second comment: The winds transfer momentum to the ice cover, and drive sea-ice drift and deformation (in part to the underlying ocean currents). However, this force is counterbalanced by the internal ice stresses and ice-ocean drag (Uotila, 2001; Leppäranta, 2011). Coriolis force also comes into effect when ice is in motion. For this reason, the ice drift deviates to the left of the wind direction in the Southern Hemisphere. Therefore, it is wind velocity that is driving ice motion. In this study, we use the ERA5 reanalysis 10-m wind components since it is the standard and most commonly used in field of research (e.g. Vichi et al., 2019; King et al., 2022; Alberello et al., 2020; Womack et al., 2022; Shokr et al., 2020). The ice motion that is measured by the buoys is already the rotated velocity, due to the Coriolis effect. Therefore, using least squares regression, we compute this deviation angle (Table 2, page 17). This deviation angle varies between the seasons due to the proximity of each array to the steering influence of the ACC. This has been clarified in Section 5.1 of the revised manuscript, lines 743-746.*

**Page 14, L414-420:** Please provide quantitative evidence when referencing "good" correlations.

*Answer: Thank you. At the end of this sentence we cross reference to Table 2 (page 17) which provides all parameters describing wind and oceanic forcing on sea ice drift. This includes the wind factor, turning angle, vector correlation, Pearson correlation and residual currents.*

**Page 15, L430-435:** What physical explanation is there for different turning angles (up the 3x the rotation) at comparable latitudes? Could this be related to the fact that you are not allowing for time variation in the turning angle and your buoys changed between predominantly meridional and zonal transport?

*Answer: We thank the reviewer for this comment. The differences between the Winter buoys were due to the proximity to the ice edge and hence the steering influence of the ACC, as discussed in lines 689-692. In spring, the ice floes exhibited a larger turning angle due to the presence of inertial oscillations, where the floes drifted less with wind forcing (lines 651-653). The latter has been related to the presence of inertial oscillations and has been further clarified in the revised manuscript, see lines 743-746.*

**Page 15, L436:** As these are linear correlation statistics, can you confirm that the variables are normally distributed? Furthermore, can you also report the p-values for these statistics, as a correlation coefficient alone is insufficient to confirm a meaningful relationship?

*Answer: Thank you for this suggestion. The variables are normally distributed. We have included histograms below of ISVP 3, in winter, and ISVP 4, in spring to confirm. All p-values for the Pearson correlations were extremely small – less than 0.05 – and thus the correlations were statically significant. This has been included in the results section (lines 596-597). A p-value however cannot be computed for the vector correlation, using the least squares regression method.*

[Figure]

*Figure: Histograms of the drift speed (left) and ERA5 wind speed (right) of ISVP 3 (in winter) and ISVP 4 (in spring), as reference for all buoys in each corresponding season.*

**Page 16, Table 2:** How do these ocean current velocities relate to what is typically reported for this region, or perhaps what is available from geostrophic velocities?

*Answer: This has been answered in the main comments.*

---

## Author Response (AR2)

***Reviewer 1 – Ruibo Lei:***

*We would like to thank the reviewer for their time and effort taken to contribute further comments and advice on our manuscript, as he has again aided in the analyses of this work. We have responded to all comments below.*

*In this document, the page numbers, line numbers, and figure numbers that we refer to in our answers relate to the revised final manuscript. All answers have been italicised.*

Review on the revised manuscript "A contrast in sea ice drift and deformation between winter and spring of 2019 in the Antarctic marginal ice zone".

The paper has been revised in detail based on the previous round of review comments, and I do not have any further serious comments. Here, I mainly propose some specific revision suggestions. It can be considered for publication after some minor revisions:

1)It is better to provide a table to summarize the intensity, their relative position and distance from the buoy, and the duration of the impact of the encountered cyclones during operations of the buoy, in order to identify the differences in the impact mechanisms of cyclones with various characteristic parameters.

***Answer:*** *We thank the reviewer for this suggestion. In our previous paper, Womack et al. (2022), we analysed the effects of individual cyclones on sea-ice drift and the wider MIZ extent. Therein, we provided a complete table (their Table 1) with corresponding figures of all passing cyclones (their Figs. 3 and 15), and how they impacted the local ice cover. Additionally, Vichi et al. (2019) provided a comprehensive investigation of the impacts that ice-landing cyclones have on the Antarctic MIZ. Herein, this manuscript rather utilizes the information, provided by these two previous studies, on cyclonic events to better understand ice drift and deformation in the context of the balance between ice type and external forcing (i.e. momentum transfers). Additionally, all previously requested information has been already added to this manuscript. Therefore, to keep the it the same length we would have to remove several components previously requested by the other reviewer. Subsequently, we also believe this would take attention away from the ice drift and deformation characteristics – a large focus of our manuscript.*

2)Line 45 "eventually consolidate into a coherent ice sheet", It would be better change to "ice cover", because the "ice sheet" is generally considered to be an ice sheet over the Antarctic continent, but not sea ice.

***Answer:*** *Thank you for this suggestion. It has been changed in the revised manuscript (line 45).*

3)Line 50 "where waves can freely propagate"--In fact, even if the sea ice has fragmented, it still has a certain dissipating effect on the waves. Therefore, waves still cannot spread freely.

***Answer:*** *Thank you for this comment. We have changed the wording to "… more freely propagate" (line 50).*

4)Line 404 "The magnitude and timing of the peaks remained the same, indicating that the relative motion in connection with the passage of the cyclones is realistic"-- I think this can indicate the high-frequency components, which correspond to periods less than 4 hours, have no significant impact on the kinematic parameters of sea ice.

***Answer:*** *Thank you for this comment. This may be true, however in the context of our study, we re-computed the meander coefficient and TSEs to show that the difference in sampling frequencies, between the ISVPs and Trident, made no significant change in the results of the drift kinematics.*

5)Line 419 "Noteworthy, are also the minor fluctuations of all dispersion components during spring" This is not a complete sentence.

*Answer:* *Thank you for this comment. We have changed this sentence to: "It is also noteworthy that there are minor fluctuations in all of the dispersion components during the spring season, ..." See lines 419-420 of the revised manuscript.*

6)Line 424 "During both seasons, increased local TSEs occurred with the passage of the cyclones, often before and/or after the cyclone core was closest to the buoys' locations" --Further explanation is needed as to why these parameters increased before the cyclone passes by.

*Answer:* *Cyclones in the Antarctic travel eastwards within the low-pressure belt, known as the circumpolar pressure trough (CPT) These ice-landing cyclones can cause rapid variations of the sea-ice cover. Amongst several other processes, they push the ice edge southwards as warm is advected poleward on their eastern flank, while also accessing a pool of cold air from the Antarctic continent on their western flank (Vichi et al., 2019). This results in the compression of sea ice, and the southward movement of the ice edge, which is then followed by the relaxation and northwards movement of the ice edge after the cyclone, when southerly winds prevail. Subsequently, the ice edge is re-arranged in a clockwise direction (Vichi et al., 2019). We have included a summary of this in lines 425-427 of the revised manuscript.*

7)Line 426 "TSEs allow to further distinguish the compression and the stretching, with the latter always observed after the former". --Further explanation is also needed why the stretching always occurred after the compression.

*Answer:* *This has been answered in question 6.*

8)Line 454 "allowing for opening of leads and rafting of floes"-- ice rafting only appears when the ice is very thin.

*Answer:* *We agree with this statement, and as reported in lines 649-652, heterogeneous (pancake-ice) conditions like the ones observed during deployment persisted during the 2019 winter period. Therefore, the ice cover remained young ice ($\approx 40$ cm thick; line 147), which raft under the action of winds and waves.*

9)Line 483 "The Winter buoys exhibited high wind factors ranging between 3.16% and 3.78%, with small turning angles ranging between -7.89° and -11.19°"-- These numbers only need to be retained to one decimal place.

*Answer:* *We thank the reviewer for this comment. However, we would prefer to keep them to two decimal places, since it does provide increased accuracy. Additionally, all of our other parameters (e.g. inertial period and correlations) and measurements from our previous manuscript (Womack et al., 2022), are provided to two decimal places. Therefore, this will keep consistency within the manuscript. If needed, readers also will be able to round up to one decimal place for their preferences.*

10)Line 539 "We attribute this to the long period of high pressure between the 7-30 October (Fig. A1a), when strong winds persisted." -- Why does high pressure correspond to strong winds?

*Answer:* *During this period of high-pressure, the winds were not as erratic and variable (in speed and direction) as during the passage of a cyclone. Thus, with more persistent wind speed and direction, there was no generation of inertial oscillations, and the power continued to remain at the lower frequencies. This would be due to the direct transfer of momentum from the winds to the sea ice i.e. no generation of inertial oscillations.*